# Diazotrophy as the main driver of the oligotrophy gradient in the Western Tropical South Pacific Ocean : results from a one-dimensional biogeochemical-physical coupled model

Audrey Gimenez[1], Melika Baklouti[1], Thibaut Wagener[1], and Thierry Moutin[1]

[1]Aix Marseille Univ., CNRS, Université de Toulon, IRD, OSU Pythéas, Mediterranean Institute of Oceanography (MIO), UM 110, 13288, Marseille, France

**Correspondence:** A. Gimenez (audrey.gimenez@mio.osupytheas.fr) or M. Baklouti (melika.baklouti@mio.osupytheas.fr)

**Abstract.** The Oligotrophy to UlTra-oligotrophy PACific Experiment (OUTPACE) cruise took place in the Western Tropical South Pacific (WTSP) during the austral summer (March-April 2015). The aim of the OUTPACE project was to investigate a longitudinal gradient of biological and biogeochemical features in the WTSP, and especially the role of $N_2$ fixation on the C, N, P cycles. Two contrasted regions were considered in this study : the Western Melanesian Archipelago (WMA), characterized by high $N_2$ fixation rates, significant surface production and low dissolved inorganic phosphorus (DIP) concentrations, and the south Pacific gyre (WGY), characterized by very low $N_2$ fixation rates, surface production and high DIP concentrations. Since physical forcings and mixed layer dynamics in both regions were similar, it was considered that the gradient of oligotrophy observed *in situ* between WMA and WGY was not explained by differences in physical processes but rather by differences in biogeochemical processes. A one-dimensional physical – biogeochemical coupled model was used to investigate the role of $N_2$ fixation in the WTSP by running two identical simulations, only differing by the presence ($\text{sim}^{WMA}$) or absence ($\text{sim}^{WGY}$) of diazotrophs. We evidenced that the nitracline and the phosphacline had to be respectively deeper and shallower than the Mixed-layer Depth (MLD) to bring N-depleted and P-repleted waters to the surface during winter mixing, thereby creating favorable conditions for the development of diazotrophs. We also concluded that a preferential regeneration of the detrital phosphorus (P) matter was necessary to obtain this gap between the nitracline and the phosphacline depths, as the nutricline depths significantly depend on the regeneration of organic matter in the water column. Moreover, the model enabled us to highlight the presence of seasonal variations in primary production and P availability in the upper surface waters in simWMA, where diazotrophs provided a new source of nitrogen (N) to the ecosystem, whereas no seasonal variations were obtained in simWGY, in absence of diazotrophs. These main results emphasized the fact that surface production dynamics in the WTSP is based on a complex and sensitive system which depends on one hand on physical processes (vertical mixing, sinking of detrital particles), and on the other hand on biogeochemical processes ($N_2$ fixation, remineralization).

## 1 Introduction

The efficiency of the oceanic carbon (C) sequestration depends upon a complex balance between the organic matter production in the euphotic zone and its remineralization in both the epipelagic and mesopelagic zones. The growth of autotroph organisms,

and therefore the assimilation of $CO_2$, is strongly linked to the nutrients' availability in the ocean surface layer (de Baar, 1994). Although nitrate ($NO_3^-$) and ammonium ($NH_4^+$) are the two main N sources taken up by autotrophs, their concentrations remain very low in the oligotrophic ocean and often growth-limiting in most of the open ocean euphotic layer (Falkowski et al., 1998). In contrast to $NO_3^-$ and $NH_4^+$, the dissolved dinitrogen ($N_2$) gas in seawater is very abundant in the euphotic

zone and could be considered as an inexhaustible N source for the marine ecosystems. Some prokaryotic organisms (Bacteria, Cyanobacteria, Archaea), commonly called diazotrophs or 'N$_2$-fixers', are able to use this gaseous N source by converting it into a usable form (i.e. $NH_3$) due to the nitrogenase enzyme system (Zehr and McReynolds, 1989; Zehr and Turner, 2001). In addition to providing a new source of nitrogen for themselves, diazotrophs release a fraction of the fixed N in the dissolved pool under the form of $NH_4^+$ and dissolved organic N (DON) in the surface waters (Bronk and Ward, 2000; Mulholland et al.,

2004, 2006; Benavides et al., 2013; Berthelot et al., 2015) and thus contribute to sustaining life and potentially C export. This new N input would seem to bring a positive advantage to the C biological pump since it would reduce the N limitation for the phytoplankton and thus enhance primary production in oligotrophic regions. However, even if diazotrophs are not limited by atmospheric $N_2$, their growth is controlled by other factors, including the availability of dissolved iron (DFe) and dissolved inorganic phosphate (DIP) (Moutin et al., 2005; Karl and Letelier, 2008). Moreover, chemostat experiments have highlighted

that $N_2$ fixation activity was highly dependent on the circadian clock and that the success of non-diazotrophs and diazotrophs depend on the interplay between light intensity and DIN concentration, and the competition for those resources (Rabouille et al., 2006; Agawin et al., 2007). In a synthesis paper, Gruber (2004) reminds that over the last decades, the work on $N_2$ fixation and the diversity of diazotroph organisms has shown a significant contribution of $N_2$ fixation to primary production in the global ocean (Falkowski, 1997; Gruber and Sarmiento, 1997; Capone et al., 1997; Karl et al., 2002), thereby calling into

question the classical paradigm of the N limitation in the open ocean (Zehr and Kudela, 2011).

   Furthermore, in the current context of climate change, Polovina et al. (2008) showed that global warming would intensify the stratification of surface waters in tropical and subtropical oceans, further reducing nutrient concentrations in the euphotic layer. It is therefore crucial to study in detail the coupling of the biogenic element cycles in the oligotrophic regions to better understand all the interactions between the processes involved in the surface production and therefore in the C biological pump.

The Oligotrophy to UlTra-oligotrophy PACific Experiment (OUTPACE) cruise has as its main objective to study how production, mineralization and export of organic matter, and associated biogenic elements C, N, P biogeochemical cycles, depend on the $N_2$ fixation process. Along the transect covered during the OUTPACE cruise, a longitudinal gradient of DIP availability was observed from low concentrations in the Melanesian Archipelago (MA) to higher concentrations in the South Pacific Gyre (SPG) (Moutin et al., 2018), closely related to an opposite gradient of primary production (Van Wambeke et al., 2018) and $N_2$

fixation rates (Bonnet et al., 2017; Caffin et al., 2018). In the framework of the OUTPACE study, it appeared crucial to investigate in detail the role of $N_2$ fixation in the surface production, using a modeling approach combining a 3D modeling study at regional scale (Dutheil et al., 2018) and a process-focused study using a one-dimensional model (this work), with the aim of explaining the contrasted ecosystems and biogeochemical cycles observed in the WTSP. Since experimental studies highlighted the significant contribution of $N_2$ fixation as a new source of N for planktonic ecosystems in the surface layer (Martínez

et al., 1983; Karl et al., 1997; Capone et al., 2005), the process of diazotrophy associated (or not) with explicitly-represented

diazotroph organisms has been implemented in numerous biogeochemical models in the last decades. As a result, more and more modeling studies have been investigating the role of diazotrophy at global scale (Moore et al., 2002, 2004; Monteiro et al., 2011), at regional scale (Coles and Hood, 2007; Zamora et al., 2010), at local scale (Fennel et al., 2002; Gimenez et al., 2016) or more specifically at population scale (Rabouille et al., 2006; Grimaud et al., 2013). While three-dimensional (3D) models

provide a general view of the studied ecosystems, computational costs often restrict the spatial and temporal resolutions and/or the complexity of the biogeochemical model. By contrast, one-dimensional models only provide a local view, but enable an accurate study of the biogeochemical processes deconvoluted from horizontal marine dynamics, at physiological (days) and ecological (months to years) time scales. In this work, we used a one-dimensional physical-biogeochemical coupled model to simulate the dynamics of the complex ecosystems observed during the OUTPACE cruise, and built two simulations to repre-

sent each of two highly contrasted regions sampled during the OUTPACE cruise, namely the Western Melanesian Archipelago (WMA) and the Western South Pacific Gyre (WGY) (see Figure 2). One of these simulations was run with diazotrophy as a proxy of the WMA region, and the second without diazotrophy as a proxy of the WGY region, to implicitly take into account the role of DFe allowing $N_2$ fixation in the MA but preventing it in the gyre (Moutin et al., 2008; Bonnet et al., 2017). The purpose of this study is to investigate the direct and/or indirect role of $N_2$ fixation in surface planktonic production and biogeo-

chemical C, N, P cycles, with the aim of determining whether the main biogeochemical differences observed in the MA and in the SPG areas can be explained or not by diazotrophy.

## 2    Methods

### 2.1    Strategy of the OUTPACE cruise and of the modeling study

The OUTPACE cruise was carried out between 18 February and 3 April 2015 from Noumea (New Caledonia) to Papeete

(French Polynesia) in the western tropical South Pacific (WTSP) (Figure 2). Two types of stations were sampled : fifteen short-duration (SD) stations dedicated to the study of the longitudinal variations of biodiversity and biogeochemistry, and three long-duration (LD) stations where Lagrangian experiments and several additional measurements (such as measurements on the settling of organic matter using sediment traps) were carried out during 6 days. The details of all the operations conducted at the different stations are summarized in Moutin et al. (2017), with a focus on the Lagrangian strategy followed at the LD

stations in de Verneil et al. (2017). Along the eastward transect from the MA to the SPG, three areas were considered regarding their different biogeochemical characteristics: the western MA (WMA), the eastern MA (EMA) and the western gyre (WGY) waters Moutin et al. (2018). In this study, we focused on the comparison of the two most contrasted areas, namely the WMA and the WGY (Figure 2). While both WMA and WGY present extremely low nitrate concentrations in the photic layer, the WMA presents higher surface production, higher $N_2$ fixation rates and lower phosphate concentrations than WGY.

As already mentioned, in order to investigate the role of $N_2$ fixation in the WTSP, we ran two identical simulations, one including the process of diazotrophy, hereafter named 'simWMA', and the second without this process, hereafter named 'simWGY'. Except for the process of diazotrophy, the two simulations were strictly identical regarding the atmospheric forcings, the initial conditions, the model formulation and the parameter values. The assumption made by using a unique set of

atmospheric forcings for two regions significantly far away is first based on the *in situ* climatological data reported in Moutin et al. (2018). These authors showed that the vertical dynamics of the water column, and especially the depths of the mixed layer were similar throughout the year in all the WTSP (see Figure SM1 in supplementary material). In addition, the atmospheric forcings calculated by the WRF atmospheric model at WMA and WGY were also very similar (see Figure SM2 in supplementary material). Furthermore, we also compared two simulations ran with the respective atmospheric forcings calculated at WMA and WGY and did not observe any significant difference, nor in the water column dynamics, neither on the biogeochemical cycles.

The methods used to measure dissolved inorganic nitrogen (DIN), DIP, $N_2$ fixation ($N_2$ fix), chlorophyll a (Chl a), primary production (PP) and particulate organic carbon (POC), as well as the corresponding data, are fully described in the companion paper by Moutin et al. (2018). For ease of reading, the following abbreviations will be used: for a given variable "X", abbreviations $X^{simWMA}$ and $X^{simWGY}$ will be used for the model outputs, respectively with and without diazotrophy, and $X^{obsWMA}$ and $X^{obsWGY}$ for the experimental data measured at WMA and WGY, respectively.

Model outputs were compared to the observations gathered during the OUTPACE cruise at WMA and WGY. For each profile presented in the Results section, we plotted the discrete values of the data collected at WMA and WGY (circles), and the average over the respective sampling periods of WMA and WGY for the model results (from 02-21-2015 to 03-02-2015 for simWMA, and from 03-21-2015 to 03-31-2015 for simWGY). Both simulations were run over ten years. Since a cyclic steady-state was reached in the near-surface layer after three years, the vertical profiles of the third year of simulation (solid line) are presented for both simWMA and simWGY. Moreover, since the outputs of simWMA provided interesting information regarding the role of diazotrophs in fueling the system with new N inputs, the ten vertical profiles of the ten-year run are all presented in the Results section.

## 2.2 The biogeochemical model

The biogeochemical model implemented in this work is embedded in the modular numerical tool Eco3M (Baklouti et al., 2006). It was based on the Eco3M-MED model (Alekseenko et al., 2014) to which two diazotrophs were added for studying $N_2$ fixation fate in the frame of a mesocosm experiment in the Noumea lagoon (Gimenez et al., 2016). For the present study, and in order to improve the model, some features of the original model presented in Gimenez et al. (2016) were modified and some new features were introduced (see Section 2.2.2).

### 2.2.1 General backgrounds

The model includes eight Planktonic Functional Types (PFT): four autotrophs (a large and a small classic phytoplankton and a large and a small nitrogen fixer), three grazers (zooplankton) and one decomposer (heterotrophic bacteria). Each of them is represented in terms of several concentrations (C, N, P and chlorophyll for phytoplankton) and an abundance (cells or individuals per liter) (Mauriac et al., 2011). Each PFT is represented by emblematic organisms indicated in brackets, and for ease of reading, living compartments are abbreviated as follows: TRI for the large diazotrophs (*Trichodesmium* sp.), UCYN for the small diazotrophs (unicellular nitrogen fixers), PHYS for the small autotrophs (pico- and nanophytoplankton), PHYL

for the large autotrophs (diatoms), HNF for nanozooplankton (heteronanoflagellates), CIL for microzooplankton (cilliates) and COP for mesozooplankton (copepods). For all the non-diazotrophic features and in agreement with literature (e.g. Luo et al. (2012)), it has been considered that a *Trichodesmium* trichome was equivalent to 100 PHYL cells and that a UCYN cell was equivalent to a PHYS cell. Yet, the conversion factor of 100 between TRI and PHYL was only applied for extensive parameters, i.e. those depending on biomass. Intensive parameters were set equal to those of PHYL, except for the specific growth rate which was instead averaged from literature since it has been experimentally demonstrated that it was lower than that of PHYL (Mulholland and Bernhardt, 2005; Hutchins et al., 2007). Parameter values, whether new or differing from those of Alekseenko et al. (2014), are given in SM: table 1.

In the model, N$_2$ fixation rates depend on the nitrogenase enzyme activity (Nase) (Rabouille et al., 2006; Gimenez et al., 2016); nitrogen fixation is the result of a balance between the increase and the decrease of the enzyme activity, which is controlled by the intracellular content in C and N, and by the NO$_3^-$ concentration. The more the cell is deprived of nitrogen, the more the nitrogenase activity is enhanced, but under the control of the intracellular C content which plays the role of "energy regulator", being itself tightly linked to the daily light cycle (Rabouille et al., 2006). Further details regarding the implementation of diazotrophy in the model are available in Gimenez et al. (2016). All the compartments and fluxes implemented in the model are summarized in Figure 1.

### 2.2.2 New features of the model

Since the Gimenez et al. (2016) modelling study was focusing on a mesocosm experiment, the assessment of the model skills was incomplete. With the new set of data provided by the OUTPACE cruise, some features of the original model were improved and some new features were introduced to correct the model major flaws or to add some realism to the model. To improve the representation of the nutricline depths which depend on the sinking of the detrital organic matter and its mineralization in the water column, we included two size classes of detrital matter associated with two different sinking rates, while the previous version (Gimenez et al., 2016) only included a single compartment of detrital material (see Table 1 in supplementary material). The large detrital particles (DETL) are fueled by the death of COP, their fecal pellets and by the quadratic mortality of PHYL. The small detrital particles (DETS) are fueled by the hydrolysis of DETL and by the linear mortality of PHYL, TRI and CIL, whereas the mortality of PHYS, UCYN, HNF and BAC fills the compartment of dissolved organic matter (DOM). The sinking rates for DETS and DETL are 1 m.d$^{-1}$ and 25.0 m.d$^{-1}$, respectively.

While several studies have shown that the intracellular C:N:P ratios in heterotrophic bacteria tend to be below Redfield values as they were enriched in N and P (Bratbak, 1985; Goldman and Dennett, 2000; Vrede et al., 2002), more recent studies suggest that these ratios could be higher than 50:10:1 and highly variables in response to physical, chemical and physiological conditions (Cotner et al., 2010; Martiny et al., 2013; Zimmerman et al., 2014). This led us to replace the 50:10:1 ratios used so far in the model for bacteria and HNF by the Redfield 106:16:1 ratio as for the other PFTs represented in the model. It is reminded however that the PFT's stoichiometry is flexible in the model and that the Redfield ratios are only used to link together the limits of the ranges of C, N and P intracellular quotas, (i.e. $Q_C^{min} = 106\,Q_P^{min}$, $Q_C^{max} = 106\,Q_P^{max}$, see Table 1 in supplementary material) thereby allowing a large variety of possible C:N:P ratios in PFTs.

Moreover, in this oligo – to ultraoligotrophic region, the regeneration of the organic matter is crucial to maintain the ecosystem balance, and certain modifications have been made in this regard : 1) to indirectly take into account the enhanced consumption of organic P through the activity of extracellular alkaline phosphatase produced by bacteria in oligotrophic areas (Perry, 1972, 1976; Vidal et al., 2003), all the half-saturation constants (Ks) for the DOP uptake were divided by one order of magnitude, 2) the hydrolysis rate of the particulate organic P was modified (from 0.4 to 2.0 $d^{-1}$) to increase the regeneration of P compared to C and N in this P-depleted area (detailed in Section 4.2.2).

## 2.3 One-dimensional physical model and forcings

The biogeochemical model has been coupled with the one-dimensional physical model described in Gaspar (1988). This model solves the conservation equations for heat, salinity, momentum and kinetic energy. The grid cell is 5 m high from surface to 200 m, and 40 m high from 200 to 2000 m. It uses a simple eddy kinetic energy parametrization with a turbulence closure scheme, resolved by the turbulent kinetic energy (TKE) equation (Gaspar et al., 1990).

The atmospheric forcings (i.e. the sensible and latent heat fluxes, the short and long wave radiation and the wind stress) for the physical model were provided by the Weather Research Forecast (WRF) model (Shamarock et al., 2008), with a spatial resolution of 15 km and a time step of 6 hours. Boundary conditions for the WRF model are provided by the American Global Forecast System (GFS) model (National Center for Environmental Prediction/National Center Environmental Prediction - NCAR / NCEP) analyzes. These analyzes correspond to a correction of the forecast using a larger number of observations during the data assimilation cycle. The WRF model is forced every 6 hours by analyzes during the processing. Only a single year of atmospheric forcing has been extracted (from September 2014 to August 2015), which was applied on a cyclical basis during the ten-year simulation. This one-year period has been arbitrarily chosen so as to cover the period from the winter mixing preceding the OUTPACE cruise to the next winter. The comparison between some physical outputs (i.e. surface temperature, surface density, mixed layer depth) and climatological *in situ* observations allowed to ensure that the one-dimensional physical model was relevant to address our scientific question (see Figure SM3 in supplementary material).

## 2.4 Initialization for the one-dimensional coupled physical-biogeochemical model

The initial profiles of temperature (T), salinity (S), DIN and DIP were constructed by interpolating mean field data from the WOA13 climatology database (Locarnini et al., 2013; Zweng et al., 2013) at the exact location of WMA (19° 13.00 S 164° 29.40 W). Initial dissolved organic matter concentrations, BAC and autotroph abundances were obtained from the vertical profiles measured in WMA: due to the homogeneity of the surface layer caused by winter mixing, the 0-70 m mean value was applied on the 0-70m layer of the initial vertical profiles. Below 70 m, initial concentrations were the same as data. Since the model includes variable stoichiometry for organisms, initial intracellular contents of non-diazotroph organisms were set to 50 %, 25 % and 75 % of their respective intracellular quota ranges in C, N and P. While there was no difference in initial C and P intracellular contents between diazotroph and non-diazotroph organisms, initial N intracellular contents of diazotrophs were set up to 50 % to take into account their metabolic advantage of fixing $N_2$. Initial concentrations of detrital compartments are nil. Initial zooplankton abundances were obtained from the BAC abundances using a BAC:HNF:CIL = 1000:100:1 ratio. In

simWGY where diazotrophs are removed, initial abundances and biomasses of TRI and UCYN were respectively transferred in PHYL and PHYS compartments, in order to strictly preserve the same initial biomasses and abundances in the two simulations.

## 3    Results

### 3.1    Vertical dynamics of the main biogeochemical stocks and flux

#### 3.1.1    Nutrients availability and $N_2$ fixation

DIN and DIP concentrations are presented in Figure 3 a) and  3 b). Strictly, DIP is the sum of orthophosphates (i.e. DIP = $[H_3PO_4]$ + $[H_2PO_4^-]$ + $[HPO_4^{2-}]$ + $[PO_4^{3-}]$), and DIN is the sum of nitrate, nitrite and ammonium (DIN = $[NO_3^-]$ + $[NO_2^-]$ + $[NH_4^+]$). However, since $[NO_2^-]$ and $[NH_4^+]$ were negligible compared to $[NO_3^-]$, DIN was assimilated to $[NO_3^-]$. From the surface to 70 m depth, $DIN^{obsWMA}$ and $DIN^{obsWGY}$ are below the quantification limit (i.e. 0.05 $\mu$M). $DIN^{simWMA}$

and $DIN^{simWGY}$ do not show any significant difference in the surface layer and range from 0.02 to 0.04 $\mu$M and from 0.03 to 0.04 $\mu$M for $DIN^{simWMA}$ and $DIN^{simWGY}$, respectively. Even if the concentrations of DIP are low in the surface layer (below 0.2 $\mu$M), some differences can be seen between WMA and WGY for both model outputs and data. Figure 3, b) shows a concentration around 0.2 $\mu$M for $DIP^{obsWGY}$ from the surface to 120 m, whereas $DIP^{obsWMA}$ is very low, with values below the quantification limit (0.02 $\mu$M) at the subsurface, with a steady increase up to 0.5 $\mu$M at 300 m depth. Regarding the model

outputs, $DIP^{simWMA}$ is close to zero from the surface to 60 m, and reaches a value of 0.70 $\mu$M at 300 m depth. $DIP^{simWGY}$ is significantly higher than $DIP^{simWMA}$ in the upper layer, with a homogeneous concentration of 0.16 $\mu$M from the surface to 175 m depth, and then increases slightly up to 0.7 $\mu$M at 300 m.

The vertical profiles of $DIN^{simWMA}$ and $DIP^{simWMA}$ show a deeper nitracline (around 75 m depth) than phosphacline (around 60 m depth), as observed with data.  Note that to reproduce this discrepancy within the model, it has been necessary

to introduce a preferential regeneration of the detrital matter in P compared to C and N in the biogeochemical model, a point thereafter detailed in Section 4.2.2. Regarding the WGY region, the observed and simulated nitracline and phosphacline are deeper than in the WMA region. The simulated nutriclines are however deeper than those measured (around 140 m and 125 m for the simWGY nitracline and phosphacline depths, respectively). While the simulated and the measured nitracline depths are both around 75 m at WMA, $DIN^{simWMA}$ is higher than $DIN^{obsWMA}$ below the nitracline. There is indeed a regular

accumulation of $DIN^{simWMA}$ below the photic zone during the ten-year simulation (Figure 3, a)), reaching at the end a high concentration of 17 $\mu$M that is not observed in $DIN^{obsWMA}$. Even if we may also note a slight variation in the phosphacline over time in simWMA, it is much less significant than the above-mentioned change in the simulated nitracline.

The $N_2$ fixation rates ($N_2$ fix) measured at WMA and WGY and the vertical profiles of $N_2fix^{simWMA}$ are presented in Figure 3 c). At the surface, $N_2 fix^{obsWMA}$ ranges from 9.0 to 30.0 nmolN.$L^{-1}$.$d^{-1}$, with a maximum rate of 35.0 nmolN.$L^{-1}$.$d^{-1}$

near 10 m depth. $N_2 fix^{obsWMA}$ then decreases gradually with depth to 9.0 nmolN.$L^{-1}$.$d^{-1}$ at 40 m before reaching low values below 40 m with values less than 1.5 nmolN.$L^{-1}$.$d^{-1}$ and a minimum of 0.1 nmolN.$L^{-1}$.$d^{-1}$ at 100 m. Regarding the model results of the simulation with diazotrophy, $N_2 fix^{simWMA}$ rates are consistent with data with a similar trend of higher rates

(around 16.0 nmolN.L$^{-1}$.d$^{-1}$ ) from the surface to 40 m depth, and decreasing values from 40 m to 70 m depth (down to 1.0 nmolN.L$^{-1}$.d$^{-1}$ at 70 m). At WGY, very low N$_2$ fix$^{obsWGY}$ were measured compared to N$_2$ fix$^{obsWMA}$, with a maximum rate of 2.0 nmolN.L$^{-1}$.d$^{-1}$ observed at the surface. In the simulation without diazotrophy, N$_2$ fix$^{simWGY}$ is nil (see Figure 3, c) ).

### 3.1.2 Chlorophylle a, primary production and carbon biomass

Figure 3 shows the vertical profiles of d) primary production (PP), e) chlorophyll a concentration (Chl a) and f) particulate organic carbon (POC) from the surface to 300 m depth. PP is significantly higher in WMA than in WGY, in both the experimental data and the model P$^{obsWMA}$ , even if PP$^{simWMA}$ values are close to the upper limit of the PP$^{obsWMA}$ range values. As for PP$^{obsWMA}$, PP$^{simWMA}$ slightly decreases from the surface to the bottom of the photic layer, before reaching low rates below 70 m.

The main differences between Chl a$^{obsWMA}$ and Chl a$^{obsWGY}$ lies in the depth of the deep chlorophyll maximum (DCM), which is around 75 m for Chl a$^{obsWMA}$ while the Chl a$^{obsWGY}$ DCM is deeper at around 140 m. The deepening of the DCM in WGY compared to WMA is a result also observed in simWGY and simWMA simulations. This deepening is nevertheless larger in the model with a DCM for Chl a$^{simWGY}$ located at 200 m, while the DCM for Chl a$^{simWMA}$ is shallower (around 50 m). In the model outputs, the location of the DCM is roughly located at the nutrient-limiting nutricline, i.e. at the phosphacline

depth in WMA and sim$^{WMA}$, and at the nitracline depth in WGY and sim$^{WGY}$. For both the data and the model outputs, we observe a difference in the depth of the DCM between WMA and WGY, but no significant difference in the DCM intensity between the two regions. Nevertheless, there is a noticeable difference in the DCM intensity between observations and simulations: Chl a$^{obsWMA}$ and Chl a$^{obsWGY}$ maximum values are equal to 0.3 $\mu$gChl.L$^{-1}$ while Chl a$^{simWMA}$ and Chl a$^{simWGY}$ maximum values are equal to 0.5 $\mu$gChl.L$^{-1}$.

The particulate carbon biomass (POC) presented in Figure 3, f) shows significant differences between WMA and WGY for both the data and the model results. First of all, there is a higher production of biomass at WMA close to the surface than at WGY. POC$^{obsWMA}$ is, at the maximum, 5-fold higher than POC$^{obsWGY}$ with maximum values at the surface reaching 5 $\mu$M. POC$^{obsWMA}$ then slightly decreases with depth to reach below 50 m values that are similar to those of POC$^{obsWGY}$ (around 1.5 $\mu$M). Higher simulated than measured POC values are also observed close to the surface. A maximum value of

7.5 $\mu$M for POC$^{simWMA}$ corresponding to the DCM is found at 65 m but is not observed in *in situ* data. A 2.5-fold lower and deeper maximum is also observed in POC$^{simWGY}$ just above 200 m, with a maximum concentration of 3 $\mu$M. POC$^{simWGY}$ concentrations remain very low between the surface and the deep maximum, while there is a significant POC production rate in simWMA with POC$^{simWMA}$ concentrations higher than 6.5 $\mu$M at the surface.

### 3.2 Seasonal variations

Unlike the available *in situ* data, the model can provide the time variations of all the above mentioned biogeochemical variables. The seasonal pattern of the nutrients pools, N$_2$ fixation, Chl a and POC is therefore shown over a 3-year period in order to focus on the seasonal variability. As already mentioned, the same atmospheric forcings are repeated every year, and they cover

the period between the last winter mixing period before the OUTPACE cruise (September 2014) and the next winter in August 2015.

### 3.2.1 Nutrients availability and $N_2$ fixation dynamics

The nutrients variation throughout the water column are in part related to the variations of the mixing layer depth (MLD) during the year. The seasonal variations of the MLD are plotted in Figure 4 a), b), d) and e). They clearly indicate a winter mixing beginning at the end of May leading to a maximum MLD of 70 m in August, followed by a longer stratified period from November to April, with a shallower (between 25 and 30 m) MLD compared to winter mixing. Figure 4, a) shows the DIN concentrations for simWMA from the surface to 200 m on a logarithmic scale. There is a slight variation of the nitracline around 70 m, but the concentration in the near-surface layer always remains below 3 nmol.L$^{-1}$ (nM), which is far below the quantification limit (50 nM).

Unlike DIN, DIP presents significant seasonal variations throughout the year (Figure 4, b). The concentrations are also presented on a logarithmic scale using the Redfield ratio (DIP x16) in order to easily compare DIN and DIP concentrations, with respect to the "classical" proportion of phytoplankton biological demand. During winter mixing, surface DIP$^{simWMA}$ increases from 0.6 to 2 nM, then remains quite stable until the end of February before regularly decreasing until June down to 0.6 nM. DIP$^{simWMA}$ then remains low during the stratified period until the next winter mixing in August. The phosphacline is always shallower than the nitracline in the simWMA and remains around 50 m depth.

Accounting for the huge computer memory this would require, the values of the different biogeochemical fluxes calculated by the model are not systematically saved. As a result, numerical values of fluxes are saved at a lower vertical resolution than concentrations (pools). For this reason we decided to represent the dynamics of $N_2$ fixation at the surface (averaged over the first 10 m) rather than as a function of depth, which would not have been as relevant as for the other variables. Figure 4 c) depicts the dynamics of the total $N_2$ fixation as well as the respective contributions of Trichodesimum sp. and UCYN to this flux. The total $N_2$ fixation at the surface varies from a minimum mean value of 15 nmol.L$^{-1}$.d$^{-1}$ during the stratified period to a maximum mean value of 20 nmol.L$^{-1}$.d$^{-1}$ reached between July and August, i.e. during the winter mixing. The major contributor to the $N_2$ fixation in simWMA is Trichodesimum sp., with on average, a contribution of 80 % of the total $N_2$ fixed against 20 % for the UCYN.

### 3.2.2 Seasonal variations of surface chlorophyll a and carbon biomass production

Figure 4 d) presents the Chl a dynamics from the surface to 200 m depth, and shows clear seasonal variations in the photic layer throughout a year. Between October and April, the Chl a$^{simWMA}$ is quite homogenous from surface to 70 m, with a DCM around 50 m reaching a maximum concentration of 0.5 $\mu$gChl.L$^{-1}$. From April and during the winter mixing, Chl a$^{simWMA}$ at the surface decreases rapidly, reaching concentrations below 0.15 $\mu$gChl.L$^{-1}$. During the same period, there is also a deepening of the DCM toward 80 m, associated with lower concentrations down to 0.4 $\mu$gChl.L$^{-1}$.

The production of C biomass in simWMA shows significant seasonal variations in the photic layer (Figure 4 e) ). The period of maximum C production at the surface lasts from October to February, with maximum concentrations of POC$^{simWMA}$

around 8 $\mu$M. As shown in Figure 3 f), a deep maximum peak of biomass is located at around 70 m, with concentrations close to 9 $\mu$M. Like for Chl $\underline{a}$, from the end of March and during the winter period, the surface $POC^{simWMA}$ decreases significantly to reach concentrations 2-fold lower than those obtained during the bloom (i.e. between November and February). While $POC^{simWMA}$ in the 0-50 m layer decreases during the stratified period, the deep maximum remains at the same depth, even if its intensity decreases with $POC^{simWMA}$ values at 70 m, reaching a minimum value of 7 $\mu$M at the end of July.

## 4 Discussion

The Western Tropical South Pacific (WTSP) has been recently qualified as a hotspot of $N_2$ fixation (Bonnet et al., 2017). It is hypothesized that, while flowing westward following the South Equatorial Current (SEC), the N-depleted, P-enriched waters from areas of denitrification located in the eastern Pacific meet in the western Pacific waters with sufficient iron to allow $N_2$ fixation to occur (Moutin et al., 2008; Bonnet et al., 2017). *In situ* data showed an ecosystem significantly more productive in WMA where $N_2$ fixation rates were higher than in WGY, where very low $N_2$ fixation rates were measured. These contrasted areas raised the question of whether the diazotrophy could be responsible for these differences observed between WMA and WGY, which led us to run two simulations only differing by taking into account (i.e. simWMA), or not (i.e. simWGY), the process of diazotrophy. The results of these two simulations were compared to the observations collected at the WMA and WGY areas during the OUTPACE cruise (Figure 3) in order to study the role of $N_2$ fixation on surface planktonic production and biogeochemical C, N, P cycles.

### 4.1 $N_2$ fixation, closely linked to the DIP availability, enhances the surface production

#### 4.1.1 Concomitant low DIP concentrations and high $N_2$ fixation rates

While DIN concentration remains below the quantification limit (50 nM) everywhere in the surface layer, there is a significantly higher DIP concentration in the photic layer at WGY than at WMA in both the data and the model outputs (Figure 3, b). The relatively high DIP concentration in WGY may be associated with inefficient or non-existent $N_2$ fixation in the gyre (Moutin et al., 2018). Because of the high Fe requirement of diazotrophs (Paerl et al., 1987; Rueter et al., 1990), the low Fe availability in WGY is assumed to prevent or significantly limit $N_2$ fixation in the South Pacific gyre (Moutin et al., 2008; Guieu et al., 2018; Moutin et al., 2018). By contrast, the high Fe availability in WMA is assumed to favor the growth of nitrogen fixers. Guieu et al. (2018) indeed measured high DFe concentrations in the photic layer in WMA provided by abnormally shallow hydrothermal sources (around 500 m deep) in the WTSP. Due to very low $N_2$ fixation measured in WGY (Bonnet et al., 2018), autotrophs organisms are N-limited, leading to a lower PP than in WMA, which results in a higher DIP accumulation in the photic layer since DIP is less consumed by organisms. Associated with lower DIP concentrations, higher $N_2$ fixation rates are observed in WMA in both the data and the model results (Figure 3, c)). DIP depletion in simWMA is due to the presence of nitrogen fixers since the two simulations have exactly the same vertical dynamics and differ only by the presence/absence of nitrogen fixers.

Studies on the role of $N_2$ fixation in the biogeochemistry of the Pacific Ocean have increased in number over the last decades, but the specific region of the WTSP remains patchily explored to date. Nevertheless, close to our studied area, Law et al. (2011) have observed a one-time DIP repletion in the surface layer due to a tropical cyclone which favored the upwelling of P-rich waters. On the basis of their Lagrangian strategy, they noticed a rapid consumption of this new DIP in correlation with a significant increase in the $N_2$ fixation rates over the following 9 days. At a larger temporal scale, Karl et al. (1997) also observed a correlation between a decrease in DIP (about 50 %) and a significant increase in $N_2$ fixation from 1989 to 1994, in the oligotrophic region of the subtropical North Pacific. The significant role of DIP availability in controlling $N_2$ fixation in the oligotrophic iron repleted WTSP (Van Den Broeck et al., 2004; Moutin et al., 2018) has been highlighted over the last decade (e.g. Moutin et al. (2005, 2008)), and the consistent results between the OUTPACE data and our model outputs, comparing simWMA and simWGY, reinforces this view of the biogeochemical functioning of this region.

### 4.1.2 Surface plankton productivity mainly driven by $N_2$ fixation in the WSTP

The patterns of surface production calculated by SimWMA and simWGY are consistent: (Figure 3,d) ): as for the *in situ* data, the model results show higher PP, POC and Chl a in simWMA (i.e. with diazotrophy) than in simWGY (i.e. without diazotrophy) in the first 0-50 m. $PP^{simWMA}$ is 20-fold higher than $PP^{simWGY}$ in the upper layer, in good agreement with $PP^{obsWMA}$ which is 15-fold higher than $PP^{obsWGY}$ (Figure 3, d)). Chl a concentrations never exceeding 0.5 $\mu$gChl.L$^{-1}$ are representative of oligotrophic waters. Model outputs and observations both show significantly deeper DCMs at WGY than WMA. The deepening of the DCM characterizes the transition from oligo (WMA) to ultra-oligotrophic (WGY) conditions during the OUTPACE cruise (Moutin et al., 2018). In the model outputs, the difference between the DCM depth in simWMA and simWGY is larger than in *in situ* data : the DCM depth in simWGY is 150 m deeper than that of simWMA, whereas the observed DCM depth in WMA is only 50 m deeper than that measured in WMA. The simulation without diazotrophy indeed presents a DCM at 200 m depth, on average 50 m deeper than that observed in the WGY region (Figure 3, e)). This deep DCM is consistent with the deep maximum of $POC^{simWGY}$ just above 200 m depth (Figure 3, d)). Both deep maxima are related to the nutricline depths located at 195 m and 185 m for $DIN^{simWGY}$ and $DIP^{simWGY}$, respectively.

As for the DCM, the nutriclines in simWGY are significantly deeper than those measured *in situ* in WGY. We assume that this gap is because $N_2$ fixation is totally removed in simWGY, whereas low but existing $N_2$ fixation still occurs *in situ* at WGY (Figure 3, c)). While the $N_2$ fixation rates reported in WGY were very low (Caffin et al., 2018), Stenegren et al. (2018) mention the presence of such diazotrophs from the UCYN group, whereas no *Trichodesmium* sp. were found in this region. The *in situ* planktonic ecosystem in WGY might therefore be slightly fueled by weak $N_2$ fixation, which is not the case for simWGY since diazotrophy is not allowed. The above assumption concerning the gap between data and model outputs is consistent with the fact that the measured PP, POC and Chl a in WGY are always slightly higher than in the model outputs for the surface layer (Figure 3, d), e) and f) ). To support this assumption, we ran another simulation considering only the presence of UCYN as diazotrophs in WGY. The results of this intermediate simulation (not shown) indicate low surface PP rates and POC concentrations, in agreement with those measured in WGY. Moreover, DIP is still available in the photic zone (though at concentrations lower than for simWGY) even if the calculated $N_2$ fixation rates were slightly higher than the measured ones. In

addition, the DCM (located around 150 m) and the nutriclines were shallower than in simWGY (i.e. without any diazotrophs). This simulation can thus be considered like an intermediate system between simWMA and simWGY, and confirms the close link between $N_2$ fixation fluxes and DIP availability.

In a previous study using nearly the same biogeochemical model including TRI and UCYN state variables in a one-dimensional configuration without physical coupling, Gimenez et al. (2016) highlighted the direct and indirect impact of the new N input provided by diazotrophs. By calculating the percentage of Diazotroph-Derived Nitrogen (DDN) in each model compartments, they followed the transfer of DDN throughout the entire trophic web as a function of time, and showed that after 25 days, 43 % of the DDN fixed by diazotrophs were found in non-diazotroph organisms. These results clearly showed that $N_2$ fixation had a significant indirect impact on the planktonic production by providing a new source of N for other organisms. DDN tracking inside the model compartments is associated with very high computational costs and could not be applied to the present study where simulations are run for several years (against 25 days for the previous study). However, it is worthwhile mentioning that the proportion of PFTs involved in total Chl a and PP was 80 % of TRI, 10 % of PHYS and 5 % of UCYN and PHYL, suggesting that the impact of $N_2$ fixation is rather direct (85 % of PP is realized by diazotrophs) than indirect. In our model, diazotrophs have therefore a net competitive advantage over the two other non-diazotrophic autotrophs.

## 4.2 A close link between MLD, nutricline depth and $N_2$ fixation

### 4.2.1 How can the nutricline depths influence $N_2$ fixation ?

In such oligotrophic areas, the positions of the nutriclines are crucial in controlling surface production (Behrenfeld et al., 2006; Cermeño et al., 2008) as they provide nutrients from the bottom to the photic layer. The equatorial Pacific Ocean is known for its complex hydrodynamic circulation induced by constant trade-winds, leading to significant variations of the thermocline position between the east and the west of the basin (Meyers, 1979). Trade winds also have an influence on the nutrient availability in the surface layer (Radenac and Rodier, 1996; Zhang et al., 2007).

During OUTPACE, the phosphacline (above 50 m) appeared shallower than the nitracline (about 75 m) in the WMA region (Figure 3, a) and b)). A similar shift between nitracline and phosphacline depths was observed 10° further south than our studied area, with a nitracline about 20 m deeper than the phosphacline (Law et al., 2011). In those N-depleted regions, diazotrophs may outcompete non-diazotroph organisms, using the unlimited atmospheric $N_2$ (Agawin et al., 2007; Dutkiewicz et al., 2014). However, their development also requires sufficient light intensity and other nutrients such as P and Fe, and the debate on their expected limitation or co-limitation is of great interest to the ocean biogeochemical community (Falkowski, 1997; Wu et al., 2000; Sañudo-Wilhelmy et al., 2001; Mills et al., 2004; Moutin et al., 2005; Monteiro et al., 2011).

On the basis of our model, we understood that it was crucial to take into account the nutricline depths, and that a shallower phosphacline than nitracline was needed to observe $N_2$ fixation rates in agreement with those measured *in situ* (Figure 3, c)). This led us to implement a preferential P regeneration in our model to reproduce the shift between the nitracline and the phosphacline (see Section 4.2.2 for more details). In our preliminary results (red lines in Figure 5), without this decoupling, the depths of the nitra- and phosphacline were located at the same depth (around 80 m), and below the MLD (Figure 5, a)

and b)). Each winter, mixing brought low concentrations of DIN and DIP in the euphotic layer. Low DIN concentrations are favorable for the development of $N_2$ fixers (Holl and Montoya, 2008; Agawin et al., 2007) but they were rapidly limited by DIP availability as the winter mixing did not provide enough DIP in the photic layer, leading to very low $N_2$ fixation rates (Figure 5, c)). In this configuration, primary production was N-limited and low compared to what was observed in the WMA region. The phosphacline had to be shallower (here about 25 m) than the nitracline, and above the winter MLD (70 m), to counteract DIP limitation. In this case, no DIN is brought by winter mixing in the photic layer, which favors $N_2$ fixers compared to non-fixer organisms, and sufficient DIP concentrations to support surface production until the next winter mixing. This simWMA configuration led to a significant development of $N_2$ fixers in the 0-50 m layer, dominated by *Trichodesmium* sp. (not shown), with consistent rates of $N_2$ fixation and PP rates (Figure 3, c) and d) ).

### 4.2.2 A preferential P regeneration needed to sustain $N_2$ fixation

To obtain with the model a phosphacline shallower than the nitracline, and thereby decrease DIP depletion in the surface layer, we had to decouple the regeneration of the detrital particulate N (DET-N) and the detrital particulate organic P (DET-P) by significantly increasing the remineralization rate of DET-P compared to that of DET-N and DET-C. The use of extracellular phosphoenzymes ( e.g. alkaline phosphatase, nucleotidase, polyphosphatase or phosphodiesterase) by microorganisms to regenerate DIP from dissolved organic P when DIP is depleted is well known (Perry, 1972, 1976; Vidal et al., 2003). Our model does not include the explicit phosphatase alkaline activity, but it is represented indirectly by giving direct access to DOP by autotrophs. However, this advantage was not sufficient to decrease P limitation enough and allow the growth of $N_2$ fixers so as to calculate $N_2$ fixation rates consistent with those measured in WMA. A preferential P regeneration of the particulate organic matter was required and obtained by increasing the DET-P hydrolysis rate compared to that of DET-C and DET-N. The location of the detrital matter regeneration in the water column is based on a balance between the sinking and the hydrolysis rates of the particulate organic matter. As mentioned in Section 4.2.2, the detrital matter is divided into two size fractions associated with two constant sinking rates of 1.0 and 25.0 m.d$^{-1}$ for the small and the large detrital particulate matter, respectively. Initially, the hydrolysis rates for the detrital C, N and P particulate matter were the same, and equal to 0.05 d$^{-1}$. The preferential regeneration of P was a posteriori obtained by increasing the hydrolysis rate of particulate P to 2.0 d$^{-1}$, without any change in the sinking rates. To illustrate how the discrepancy between nitracline and phosphacline depths can be attributed to preferential P regeneration, DIN and DIP concentrations and $N_2$ fixation rates in simWMA calculated with and without preferential P regeneration (i.e. preferential hydrolysis of P particulate matter) have been compared (Fig. 5). This figure shows the deepening of the phosphacline in the simulation without preferential P regeneration. More importantly, without preferential P regeneration, the phosphacline depth is deeper than the MLD at 70m, which prevents DIP input in the surface layer during winter mixing, and leads to a strong limitation of diazotrophs by P. This stronger P-limitation without preferential P regeneration can also be observed at the cellular scale by analyzing the intracellular P quota of $N_2$ fixers. In the simulation without preferential P regeneration, the relative intracellular quota of P in TRI and UCYN is in average respectively 8 and 16 times lower than with preferential P regeneration. As $N_2$ fixation alleviates N limitation for diazotrophs, their growth is thus limited by P availability. The significant decrease in relative intracellular P quotas has therefore a significant impact on their growth, and consequently

explain the lower $N_2$ fixation rates shown in Figure 5, c). This preferential P remineralization was also used by Zamora et al. (2010) who investigated different mechanisms that might be able to explain the N-excess observed in the North Atlantic main thermocline. Even if their model did not include $N_2$ fixation, they concluded that the N excess observed would be a consequence of a co-occurrence of a preferential P remineralization and a surface N input provided by $N_2$ fixation. With the same aim of studying the N excess observed in the North Atlantic main thermocline, Coles and Hood (2007) implemented a more complex model including the $N_2$ fixation process and variable stoichiometry for the non-living compartments. They concluded that a preferential P regeneration was needed to generate the N excess anomalies observed in the subsurface North Atlantic, and the preferential P regeneration was obtained by increasing the P remineralization rates relative to N. In both their and our study, the change in P remineralization rate was necessary to reduce upper surface P limitation for diazotrophs and to obtain $N_2$ fixation rates consistent with observations.

## 4.3 N from $N_2$ fixation accumulates in the main thermocline

By running the model simulations over ten years, we observed the storage of the new N input by diazotrophy. The nitrate accumulation observed in simWMA from 70 m to 500 m (Figure 3, a)), reaching concentrations of 17.0 $\mu$M after a run of ten years, is obviously overestimated as we used a one-dimensional model, without any horizontal exchange. The horizontal advection which would occur in the field is not represented here, and without any loss processes taken into account, the annual N input by $N_2$ fixation accumulates, as observed in simWMA. The interesting point is the location of this accumulation around the first 400 m of the main thermocline between 100 and 500 m depth. This result is consistent with some studies which have investigated the N excess in the ocean, using for instance the $N^\star$ tracer ($N^\star= NO_3^- - 16 \times PO_4^{3-}$, Gruber and Sarmiento (1997)) and the $N_2$ fixation contribution to this N-excess (Bates and Hansell, 2004; Hansell et al., 2004; Landolfi et al., 2008; Zamora et al., 2010). A companion paper in this special issue investigates in detail the N excess observed in the WTSP in relation with $N_2$ fixation (Fumenia et al., *under rev.*, this issue). Our model results clearly show an accumulation of N in the 100-500 m layer which results from the new N input by diazotrophy, as this is the sole external N source implemented. This accumulation constantly increases every year by an average of 449.6 mmolN.m$^{-2}$ while the annual integrated $N_2$ fixation provides 451.0 mmolN.m$^{-2}$. After benefiting the upper water ecosystem, more than 99.5 % of new N derived from $N_2$ fixation ends in the DIN pool from the 100-500 m layer. We use a one-dimensional model which is not intended to provide any quantitative conclusion regarding this N accumulation, but these calculations explain the annual DIN accumulation observed in Figure 3 a). According to the model, $N_2$ fixation may explain the N excess observed *in situ* around the main thermocline in the WTSP, as reported by Fumenia et al. (*under rev.*, this issue).

## 4.4 $N_2$ fixation leading to seasonal variations in the WTSP

To date, the WTSP, and more generally the South Pacific Ocean, is much less studied than the North Pacific Ocean which has been sampled since the late 1980s within the framework of the Long-term Oligotrophic Habitat Assessment (ALOHA) near the Hawaii islands. The South Pacific has been sampled from west to east during the BIOSOPE (Claustre et al., 2008) and OUTPACE (Moutin et al., 2017) French oceanographic cruises and many other cruises (e.g. Moisander et al. (2012), and the

cruises involved in the GLODAPv2 project, Olsen et al. (2016)) providing a spatial (Fumenia et al., *under rev.*, this issue), but not a temporal overview of the south tropical Pacific. To date, the seasonal variations were only studied during the DIAPALIS cruises (http://www.obs-vlfr.fr/proof/vt/op/ec/diapazon/dia.htm) in several stations located in the MA close to New Caledonia. By means of our one-dimensional model, we can analyze the annual variability of the entire ecosystem implemented in the model in order to corroborate, or not, certain hypotheses raised from the OUTPACE data analysis.

Seasonal variations obtained in $sim^{WMA}$ (Figure 4) allow to trace the annual "history" of the WMA region : During the winter period, vertical mixing intensifies and replenishes the surface layer in DIP but not in DIN, as the nitracline is deeper than the MLD (70 m), whereas the phosphacline is shallower (Figures 3 a) and b)). The newly available DIP in the surface layer is immediately followed by an increase in $N_2$ fixation rates in June (Figure 3, c)), which then remain quite stable until October before slightly decreasing until the next winter mixing. There is a close relationship between DIP availability and the $N_2$ fixation rates since $N_2$ fixation decreases as the DIP concentration decreases with the DIP gradual consumption after the mixing period. Because $N_2$ fixation provides a new source of N (characterized by a rapid turnover time, as it is immediately used and transferred into the ecosystem), the DIP is consumed, thereby generating seasonal variations in the surface layer. Although autotrophs are dominated by $N_2$ fixers, $N_2$ fixation benefits the entire planktonic trophic web and enhances the surface production which is directly controlled by nutrient availability. We therefore observe surface seasonal variations in Chl $\underline{a}^{simWMA}$ and $POC^{simWMA}$ , with maximum values from October to the end of March, and a less intense and deeper signal around 70 m (which corresponds to the nitracline depth from April to the end of May). During the stratified period, when $N_2$ fixation is the lowest, non-diazotroph organisms grow deeper where DIN is available. The temporal evolution of simWGY is not shown here as there is no seasonal variations associated with $N_2$ fixation, which is the focus of the study presented here. The absence of diazotrophy leads to a deepening of the nitracline and available DIN is deeper. DIP is never exhausted in the surface layer because the model implicitly assumes iron limitation to prevent $N_2$ fixation. The maximum biomass and DCM are constant throughout the year, significantly less intense than in simWMA and located near the nitracline around 200 m.

## 5  Conclusions

The purpose of this study was to investigate the direct and/or indirect role of $N_2$ fixation on surface planktonic production and biogeochemical C, N, P cycles, with the aim of determining whether the main biogeochemical differences observed in the MA and in the SPG areas could be explained or not by diazotrophy. For this purpose, a new coupled one-dimensional physical-biogeochemical model has been built based on the Eco3M-Med model. Two simulations were designed, only differing by the presence/absence of diazotrophs. They enabled us to reasonably reproduce the main biogeochemical characteristics of the two biogeochemical areas (WMA and WGY). The model could also reproduce the high contrast between the two regions, such as (i) the high/low DIP availability respectively associated with significant/ negligible $N_2$ fixation and surface production, (ii) the higher/ lower depth of the nutriclines characteristic of oligotrophic (WMA)/ultra-oligotrophic (WGY) regions, (iii) the large/small gap between DIN and DIP nutriclines and the subsequent consequences on nutrients input in the surface layer during winter mixing.

Winter mixing allows the annual replenishment of the surface layer in excess P, creating ideal conditions for diazotroph growth and intensive $N_2$ fixation. The development of diazotrophs can counteract DIN limitation for the entire planktonic trophic web in the photic layer in WMA, which leads to significant seasonal variations due to the progressive exhaustion of DIP after winter mixing. Throughout the year, we then evidenced a shift from N to P limitation of the planktonic community growth in MA. The strong influence of seasonal variations shown by the simulations in the WTSP, and generally not considered in tropical areas, needs to be further studied and backed up by *in situ* observations.

*Acknowledgements.* This is a contribution of the OUTPACE (Oligotrophy from Ultra-oligoTrophy PACific Experiment) project (https://outpace.mio.univ-amu.fr/) funded by the French research national agency (ANR-14-CE01-0007-01), the LEFE-CyBER program (CNRS-INSU), the GOPS program (IRD), the CNES, and from the European FEDER Fund under project 1166-39417. The OUTPACE cruise (http://dx.doi.org/10.17600/15000900) was managed by the MIO (OSU Institut Pytheas, AMU) from Marseilles (France). The authors thank the crew of the R/V L'Atalante for outstanding shipboard operation. G. Rougier and M. Picheral are warmly thanked for their efficient help in CTD rosette management and data processing, as is Catherine Schmechtig for the LEFE CYBER database management. The authors also acknowledge the staff of the "Cluster de calcul intensif HPC" Platform of the OSU Institut Pythéas (Aix-Marseille Université, INSU-CNRS) for providing the computing facilities Finally, the authors gratefully acknowledge M. Libes and C. Yohia from the Service Informatique de Pythéas (SIP) for technical assistance and for providing the atmospheric forcings for the present modelling study.

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

**List of Figures**

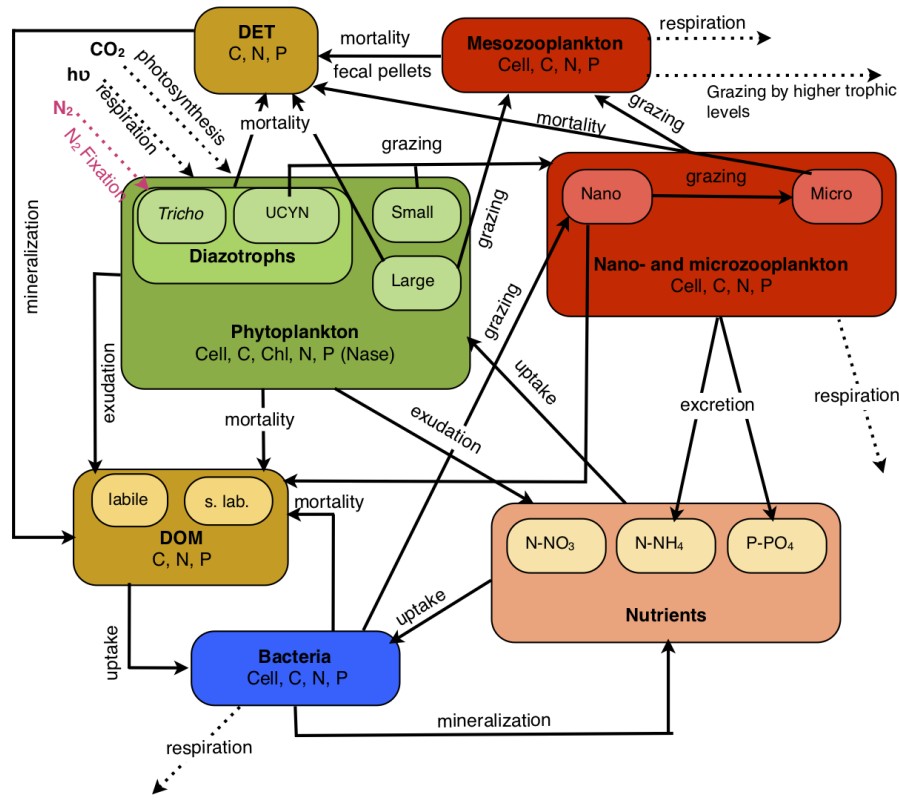

**Figure 1.** Conceptual diagram of the biogeochemical model for the physical-biogeochemical coupled 1D-vertical model used in the OUT-PACE project.

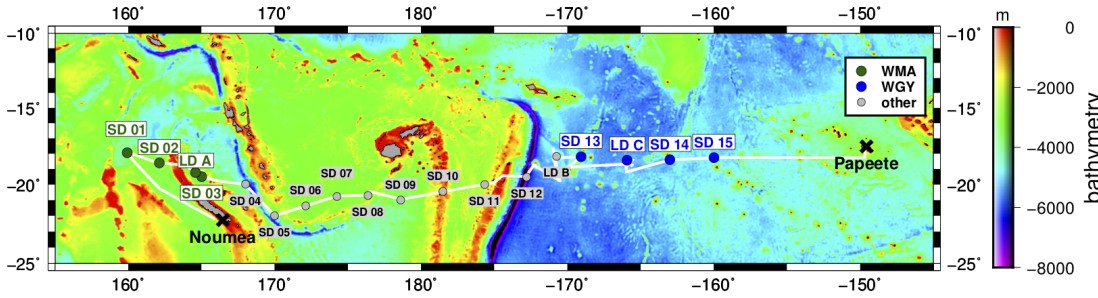

**Figure 2.** Transect of the OUTPACE cruise with the location of the short duration (SD) and long duration (LD) stations superimposed on a bathymetry map (GEBCO_2014 grid). The two main regions studied in this work are shown in green for WMA and in blue for the WGY.

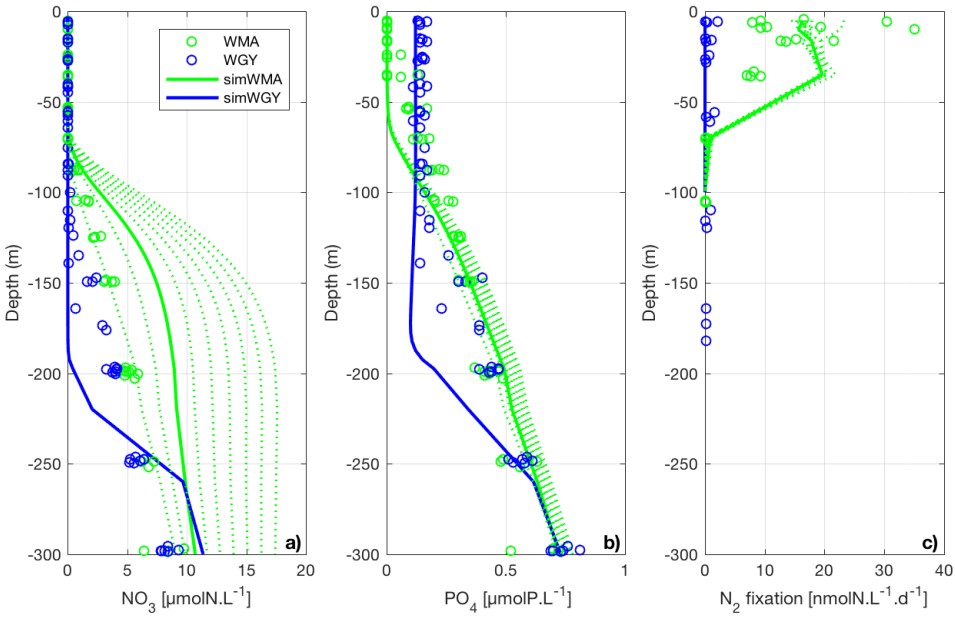

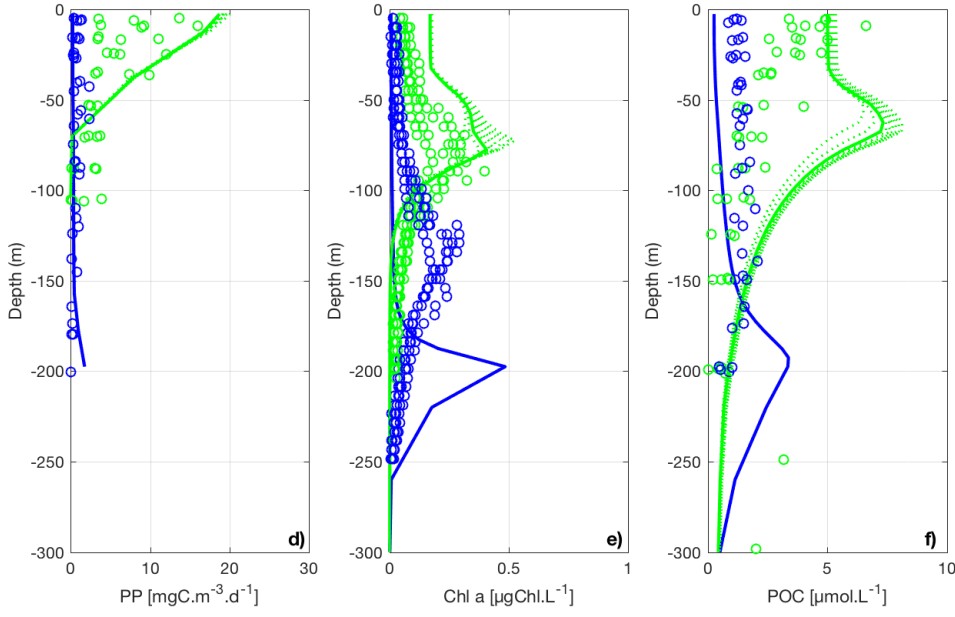

**Figure 3.** Vertical profiles of (a) dissolved inorganic nitrogen (DIN), (b) dissolved inorganic phosphorus (DIP), (c) N$_2$ fixation rates, (d) primary production (PP), (e) chlorophylle a (Chl a) and (f) particulate organic carbon (POC) during the OUTPACE cruise in the WMA region (green circles) and in the WGY region (blue circles), and from the model results of the simulation with diazotrophy as a proxy of the

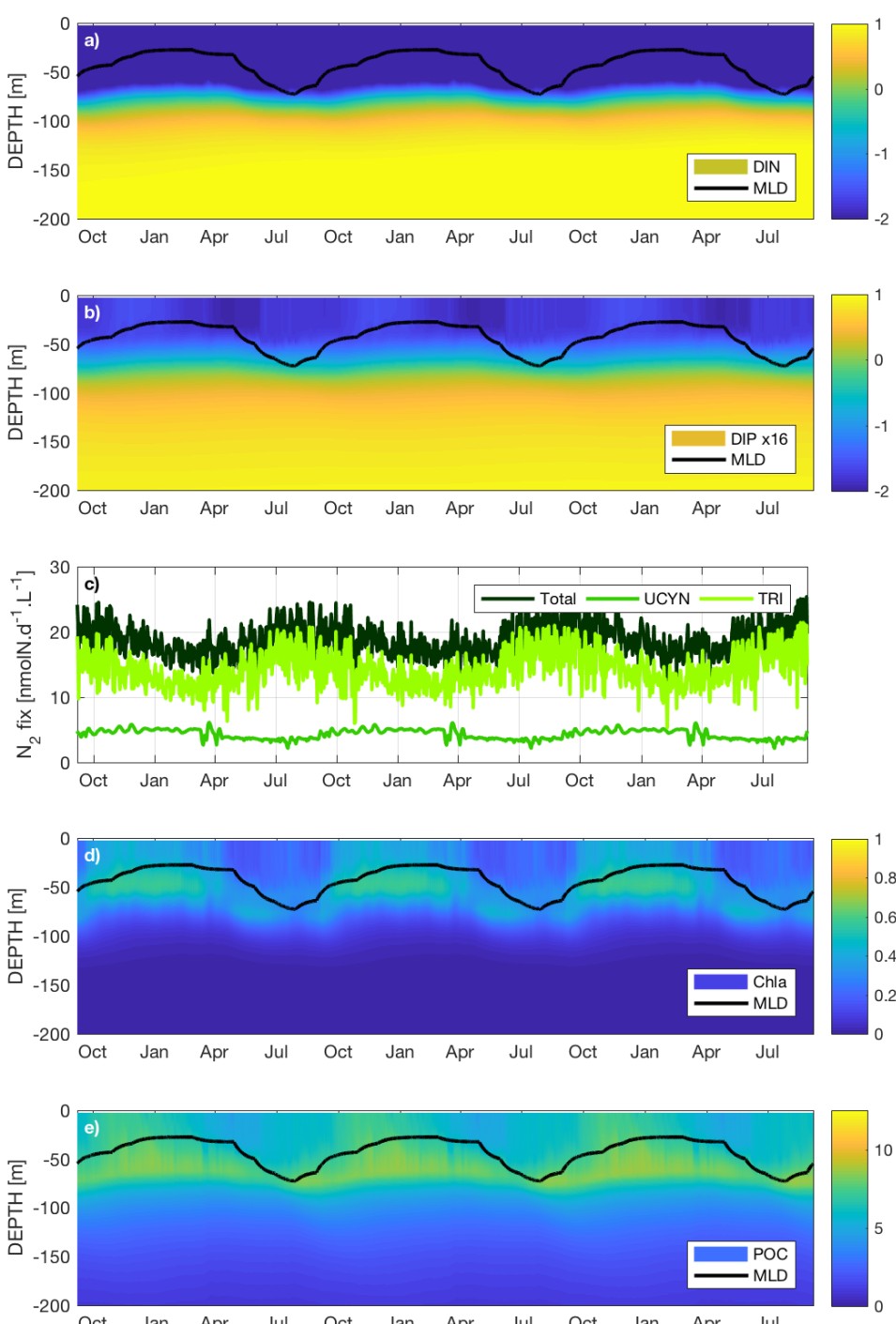

**Figure 4.** Seasonal dynamics of (a) dissolved inorganic nitrogen (DIN) on a logarithmic scale in μM, (b) dissolved inorganic phosphorus

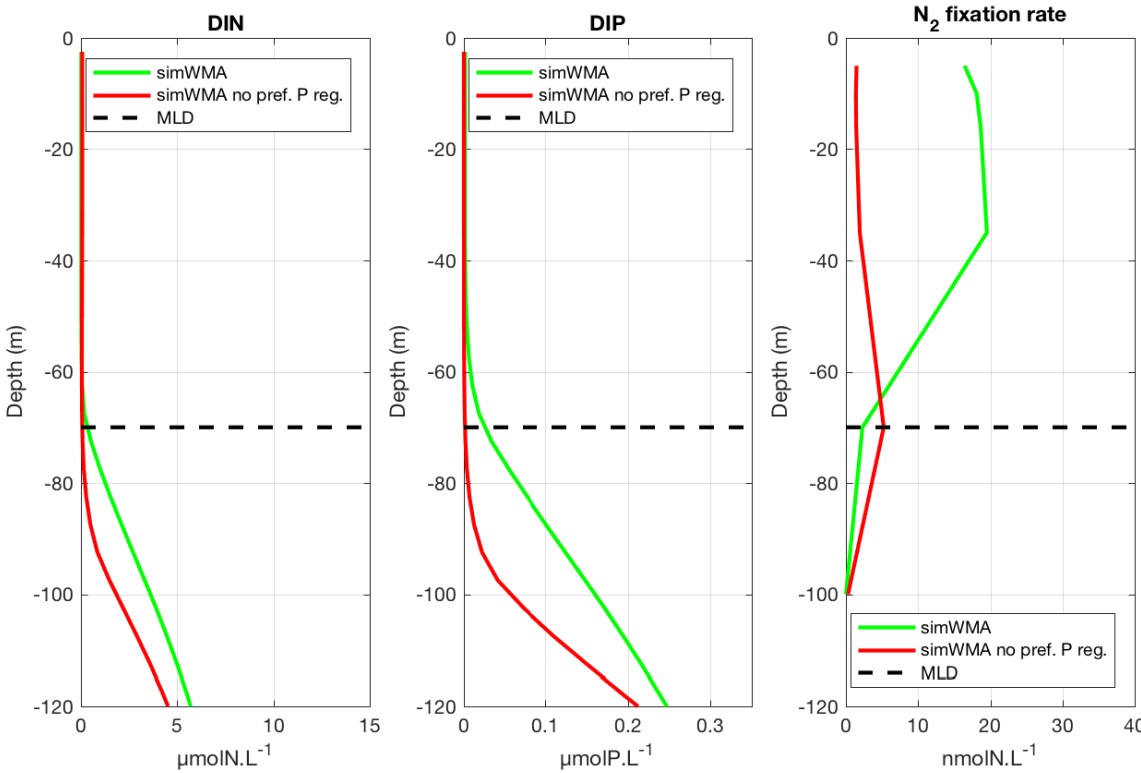

**Figure 5.** Vertical profiles of (a) dissolved inorganic nitrogen (DIN), (b) dissolved inorganic phosphorus (DIP), (c) $N_2$ fixation rates for the simulation with diazotrophy, as a proxy of the WMA region (sim$^{WMA}$) with the preferential P regeneration (in green) and without the preferential P regeneration (in red). The horizontal black dashed line represents the depth of the maximum mixing layer calculated at 70 m depth.