# Peer review of "Diazotrophy as the main driver of the oligotrophy gradient in the Western Tropical South Pacific Ocean: results from a one-dimensional biogeochemical-physical coupled model"

_Biogeosciences, 2018_

## Short Comment (SC1) · 19 Apr 2018

Dear authors,

I have read this manuscript with a great interest since the topic is higly important for the understanding of the complexity of the primary productivity controlling factors and the role of $N_2$ fixation in nutrient limited oceans.

General comments:

I find that the manuscript brings interesting combination of field research with modelling. Also, the exploration of nitracline and phosphacline depths variations in oligotrophic and ultra-oligotrophic regions of WTSP, including another analyses, enables a discussion of a high quality. Presented results are clear inspite of the complexity of the conducted research. I find the manuscript excellent, according to my skills to judge an overall scientific contribution. At the same time, I see some possibilites for improvements in Introduction, Methods and Discussion.

Topic – light
Since You mentioned benefitial role of $N_2$ fixers for the whole plankton community, which is in line with Agawin et al. (2007) and other studies which You cited, it would be interesting to explicitly include a sentence about the interplay between nutrients (dominantly DIN) and light as the common controlling factors in competition between $N_2$ fixers and non-fixers. It is obvious that Your research went further in more details concerning the link between different types of nutrient limitation and $N_2$ fixation, but for me as a reader, at least a small part about the light is missing.

I suppose that since the studied area spreads within a short range of geographic latitudes, You did not consider to include the topic of light in this particular manuscript, especially because of small differences in hours of daylight over the year in such an area close to the Equator? Therefore, You focused on the topic of nutrients without mentioning light explicitly, but only indirectly via citations?

Topic – phosphoenzymes
I suggest that You slightly improve the part about the phoshphoenyzmes that are important in usage of DOP during P-limitation (see below my suggestion). In addition, there is a fine opportunity for the inclusion of findings by Dyhrman et al. (2006) reffering to the C-P lyaze enzyme in *Trichodesmium*, which can enable this organism to get the competitive advantage over other marine phytoplankton that do not use phosphonates that are considered to contribute even up to 25% in DOP (Dyhrman et al. 2006).

Topic – stoichiometric ratios C:N:P for heterotrophs
I did not understand the reason for implementation of Redfield ratios for the heterotrophs since the literature supports molar C:N:P of e.g. 50:10:1. I see that later in the manuscript the rates of N and P mineralization are adjusted in the model, so You did some compensation, which seems to me rational to do. However, I think that many readers would like to see a sentence with explanation for the Redfield ratios for heterotrophs.

Technical details

My question marks are there only to provoke Your effort to slightly improve the presentation, rather than expecting the explicit answers during the open discussion, as far as I am concerned.

Page 1:
line 8 - only differing by the presence or absence of diazotrophs

line 13 – which seasonal changes? It seems that Abstract would be better if you write precisely here about the main results of your research regarding those seasonal changes.

Line 22 – I feel that the logics of the sentence is inadequate. The word „although" would suggest opposing statement in the second part of the sentence, but in fact the outcome is pretty logical. The area is oligotrophic and since $NH_4$ and $NO_3$ are the main inorganic N species taken up by osmotrophs, those nutrients remain low. So, I suggest this sentence: „Since nitrate ($NO_3^-$) and ammonium ($NH_4^+$) are the two main N sources taken up by autotrophs, their concentrations remain very low in the oligotrophic ocean being frequently growth-limiting factor in most of the open ocean....

Page 2:
Line 2 - Some species of prokaryotic organisms....
Line 5 – Would it be good to ammend this part with a detail about the nature of dissolved N? I mean on this „diazotrophs release dissolved inorganic and organic N..."?
Lines 7-8: I suggest „since it would reduce the N limitation for the phytoplankton and thus enhance primary production in the oligotrophic regions.
Explanation: This is not "characteristic" because some oligotrophic seas are P-limited, therefore "characteristic" seems not to be associated to N exclusively. "in the oligotrophic regions" sounds a bit better at the end of the sentence. Just a suggestion.
Line 10 - bioavailability of dissolved iron (Fe) and phosphate (P).....
Line 13 – It would be nice to have a reference at the end of this sentence since You write about classical paradigma?

After this sentence it would be nice to extend the Introduction by Agawin et al. (2007) and Rabouille et al. (2006), which You already cited, but an explicit note about the role of light for $N_2$ fixation in Your manuscript is lacking. You might briefly add here that $N_2$ fixation is highly dependent on the circadian clock (Rabouille et al. 2006) and that the success of non-diazotrophs and diazotrophs depend on the interplay between intensity of light and DIN concentration and the competition for those resources.

Line 22 – DIP availability if the statement is strictly for inorganic P.

Page 3:
Line 3 – Is it ecosystem or You explored ecosystems, one with diazotrophy and one without?
Line 20 – their different biogeochemical characteristics (only a lack of letter „i" in their)

Page 4:
Lines 3,4 – I do not see in the manuscript where did You mention earlier anything about „ten years"?
Line 4 – How did You define „near-surface layer"?
Line 14 – „consumers" could be changed to „grazers"? However, this is less important.
Line 25 – after „energy regulator" it seems appropriate to extend the sentence with „, being itself tightly linked to the daily light cylce (Rabouille et al. 2006). Just an example.

Page 5:
Line 4 – Hereafter You have numerous dots after some units, these dots should be corrected.
Lines 5,6: Why did You change C:N:P from 50:10:1 for heterotrophs, which is supported by literature (Goldman and Dennett (2000), Fagerbakke et al. (1996), Chan et al. (2012), Alekseenko et al. (2014), for which You cite 50:10:1), to the Redfield ones?
Line 9 – Hereafter You write "phosphatase alkaline". Is it nicer to write alkaline phosphatase, or it has to be phosphatase alkaline?

Line 34 – According to which reference You chose the ratios 1000:100:1? Can You extend the sentence with a comment about grazing pressure from right to the left in these abundances ratios? Then there is no need for reference maybe.

Page 7:
Line 21 – "particulate carbon biomass", "particulate organic carbon" and "C biomass" are present in the manuscript. Are some of these names associated to the same variable? Can You simplify this?
Line 32 – Why not "over ten years" if You are able to perform model for ten years?

Page 8:
Line 28 –maybe extending the sentence with "compared to winter mixing".

Page 9:
Line 26 – exclude one of the words, "new" or "fresh" because they carry the same information.

Page 10:
Line 24 – Did you provide an abbreviation explanation for PP before this point in the manuscript?

Page 11:
Line 7 - However, their development also requires sufficient intensity of light and other nutrients…
Line 26 –Why "or"? Is it better to write "e.g." than "or". There are other phosphoenzymes potentially used to get P from DOP, especially by *Trichodesmium*. I suggest "(alkaline phosphatase, nucleotidase, polyphosphatase, phosphodiesterase)".
Then You could extend the discussion by "Moreover, C-P lyase found in *Trichodesmium* (Dyhrman et al. 2006) is another enzyme that enables this organism to use previously considered non-bioavailable fraction of DOP, i.e. phosphonates that represent circa 25% of DOP (Dyhrman et al. 2006). Taking into account this possibility, *Trichodesmium* thrives in the oligotrophic oceans with a great success".

Page 12:
Lines 17-18 – "around the main thermocline between 100 and 500 m depth". Can You provide a figure for the vertical profiles of temperature?

Page 13:
Line 7 – project, Olsen et al. (2016)). Comment: add comma after "project".

Page 14:
Line 6 – nutriclines depths
Lines 8-9 – Is excess DIP ideal for growth of diazotrophs if the light intensity is not suitable? Maybe "Excess DIP in N-limited surface ocean, if supported by sufficient photosynthetically active radiation, would favor the growth of diazotrophs in comparison to non-diazotrophs"?

Page 24:
Figure 4c – Why do You use "Trichos" if You defined the abbreviation in the Methods as TRI?

References (*already cited)

        *Agawin NSR, Rabouille S, Veldhuis MJW, Servatius L, Harriët M, Van Overzee MJ, Huisman J (2007) Competition and facilitation between unicellular nitrogen-fixing cyanobacteria and non-nitrogen fixing phytoplankton species. Limnology and Oceanography 52(5): 2233–2248.

Chan et al. (2012) Transcriptional changes underlying elemental stoichiometry shifts in a marine heterotrophic bacterium. Frontiers in microbiology doi: 10.3389/fmicb.2012.00159

Dyhrman TS, Chappell PD, Haley ST, Moffett JW, Orchard ED, Waterbury JB, Webb EA (2006) Phosphonate utilization by the globally important marine diazotroph Trichodesmium. Nature 439(7072): 68–71.

Fagerbakke KM, Heldal M, Norland S (1996) Content of carbon, nitrogen, oxygen, sulphur and phosphorus in native aquatic and cultured bacteria. Aquatic Microbial Ecology (10): 15–27.

Goldman JC, Dennett MR (2000) Growth of marine bacteria in batch and continuous culture under carbon and nitrogen limitation. . Limnology and Oceanography 45(4): 789-800.

*Rabouille S, Staal M, Stal LJ, Soetaert K (2006) Modeling the dynamic regulation of nitrogen fixation in the cyanobacterium Trichodesmium sp. Applied and Environmental Microbiology 72(5): 3217–3227.

---

## Referee Comment (RC1) · Anonymous Referee #1 · 4 May 2018

**1   General comments**

The authors compare two 1D simulations only differing in the presence of diazotrophy to examine the role of N fixation for plankton production and biogoechemical cycles in observations in the Western Tropical South Pacific. While this aim of the study is given at in the introduction and the simulations presented are well suited to address this aim,

in their interpretation of the results the authors claim to show the control of preferential P regeneration (a model assumption that is not tested) on N fixation. In my opinion, the simulations necessary to justify this latter claim are not provided. The results are very interesting and relevant to the current discussion on N fixation, and the underlying processes identified seem reasonable, but either the interpretation needs rephrasing or additional simulations need to be presented to acknowledge the causality implied by the study setup. Affected sections are identified in detail below. Furthermore, some validation of the simulated physics (MLD) on the seasonal scale would be needed to give confidence in the validity of the results.

1. Does the paper address relevant scientific questions within the scope of BG?
   yes

2. Does the paper present novel concepts, ideas, tools, or data?
   yes

3. Are substantial conclusions reached?
   yes, but see below.

4. Are the scientific methods and assumptions valid and clearly outlined?
   partly. The design of the model study appears valid, but the way the interpretation of the results is phrased suggests a different cause and effect than the study design. Take, for example, the authors' claim in the abstract and the discussion that they "evidenced that the nitracline and phosphacline had to be respectively deeper and shallower than the Mixed-Layer Depth (MLD) .. [to create] ... favourable conditions for the development of diazotrophs" (p1,l8-11) and "concluded that a preferential regeneration of the detrital phosphorus (P) matter was necessary to obtain this gap between the nitracline and the phosphacline depths ..." (p1,l11-13). But neither the depth of the nutriclines nor the preferential P regeneration are manipulated in the simulations presented. Causality here

goes the other way, in my opinion: the authors set up a system with preferential P regeneration and then show how N fixation creates different biogeochemical regimes.

5. Are the results sufficient to support the interpretations and conclusions?
no. The simulations required to show the control of P availability on N fixation as claimed (e.g., additional simulations without preferential P regeneration), are mentioned in the discussion as preliminary (p11,l34 - p12,l1), but are not shown. Either the interpretation needs to change from "control of P availability on N fixation" to "effects of N fixation on nutricline and seasonality", or additional simulations need to be provided.

6. Is the description of experiments and calculations sufficiently complete and precise to allow their reproduction by fellow scientists (traceability of results)?
no. The study refers to previous studies (Gimenez et al. 2016 - G2016, Alekseenko et al. 2014 - A2014) for all equations and parameter values, stating that "for all the non-diazotrophic features, TRI are parameteriyed as 100 PHYL cells ... and UCYN as PHYS." (p4, l20-21). Yet both references differ in some of the parameter values. Yet G2016 give specific mortality rates of UCYN as $1.16 \ 10^{-6}$ $s^{-1}$ compared to $1.16 \ 10^{-7} \ s^{-1}$ for PHYS (A2014), and the max. growth rate of TRI as $2.08 \ 10^{-6} \ s^{-1}$ compared to $2.3 \ 10^{-5} \ s^{-1}$ for PHYL (A2014). This is confusing, although maybe not the fault of the authors: What is a typo, what is related to the conversion of values for TRI as 100 PHYL cells, which values are used in the present study?

7. Do the authors give proper credit to related work and clearly indicate their own new/original contribution?
yes.

8. Does the title clearly reflect the contents of the paper?
it reflects the study setup, but not the interpretation/conclusions drawn from it.

9. Does the abstract provide a concise and complete summary?
   it needs rephrasing to fit the study design

10. Is the overall presentation well structured and clear?
    yes

11. Is the language fluent and precise?
    yes, only in few sections a bit redundant

12. Are mathematical formulae, symbols, abbreviations, and units correctly defined and used?
    yes. some minor inconsistencies are pointed out below.

13. Should any parts of the paper (text, formulae, figures, tables) be clarified, reduced, combined, or eliminated?
    the abstract, discussion and conclusion should be brought in line with the simulations/results presented.

14. Are the number and quality of references appropriate?
    yes

15. Is the amount and quality of supplementary material appropriate?
    a supplement could clarify which parameter values were used (cf. no. 6 above)

**2 Specific comments**

1. p3, l24-27:

   (a) you use two simulations identical in forcing and physics to simulate two locations about 40 deg. longitude apart with different biogeochemical characteristics. Is it justified to apply the same physics and forcing to both locations?

If so, could you provide evidence for this, e.g. observations? It this choice compromising the fit between model and observations for station WGY?

(b) it confused me that the simulations were named after the different locations, e.g. simWGA, while applying the same physics/forcing. I was expecting different physics. In my words, I would say you used an idealized environment representative of station WMA to test the effect of N fixation, and got results in good agreement with the other location in the case of no N fixation. To me it would thus be more intuitive to call the simulations Nfixation and no-Nfixation or something similar.

2. p4,l20-21. How does your parameterisation of the different PFTs compare to other parameterisations of diazotrophy in the literature?

3. p4,l29-p5,l12: you modify the published biogeochemical model parameterisation substantially. Why were those changes necessary? Did you perform any optimisation or did you tune the model by hand? Was the model particularly sensitive to any of the parameters? Are the physics simulated in agreement with observations?

4. p5,l22; p8,section 3.2.1.; Fig. 4: you mention winter and winter mixing. Is this appropriate for a tropical region at 17 deg S? As winter mixing I would understand much deeper and more rapid isolated mixing events as are characteristic for the temperate latitudes. Is the gradual deepening simulated for the MLD in Fig. 4 not mostly a result of increasing salinity in the dry season?

5. p6,l18: Could you mention here already that the shallower phosphacline than nitracline results from the model setup with preferential P regeneration? This aspect is not a result of the simulations presented here and would have made it significantly easier for me to understand this section.. I acknowledge that there are different writing styles. In the case of this ms, readability of the results section

would in my opinion be greatly improved if short explanations for the model data and the model-data discrepancies that follow directly from the model assumptions (i.e., the 1D model accumulating additional N due to missing advection; the shallower phosphacline compared to nitracline because of preferential P regeneration), were given already in the results section. The discussion section would then focus on putting the findings into context/analysing more complex mechanisms.

6. Fig. 2: could you add e.g. a small inset showing the location of this map on the globe?

7. Fig. 3: you do not mention the identity of primary producers in the WMA simulation. With the confusing parameter value differences mentioned above I am not sure whether diazotrophs in these simulations only have net competitive advantages over the two other phyto types and would therefore be expected to dominate the WMA simulation? Moreover, instead of the fairly detailed description of the curves it might ease comprehension of the results to point out that in both simulations and observations the Chl a maximum is roughly located at the nutrient-limiting nutricline.

8. Fig. 4: I guess the black line is the simulated MLD. You point out the connection between the MLD and the nutriclines on p11,l1-2. So how realistic is it compared to observations? Moutin et al. 2018 (BGD) present observation-based values from de Boyer Montegut et al. 2004 showing maximum MLD in August (here October) and high Chl from April to August (here October to April). What causes this discrepancy and how does it affect your results?

9. causality between N fixation and preferential P regeneration:

   • p9,l29-p10,l2: "The ... role of DIP availability in controlling N2 fixation ... has been highlighted over the last decade ..., and the consistent results

between the OUTPACE data and our model outputs, comparing simWGA and simWGY, reinforces this view of the biogeochemical functioning in the region." - You do not test the effects of P availability in the simulations presented, you assume high/sufficient P availability and manipulate N fixation. Your simulations thus show the connection the other way round: without N fixation you have P left over.

- p11,l10-11: as above: the simulations presented do not show that a shallower phosphacline than nitracline was needed to observe N fixation rates in agreement with observations. they show that assuming N fixation leads to this gap between phospha- and nitracline. Could it be that you still had the simulations without preferential P regeneration in mind, that are mentioned on p11,l34-p12,l1?

- p11,l23-25: I have to admit during the first read it took me until here to realize how the difference in phospha- and nutricline depths came about. The preferential P regeneration is a model assumption. Why not mention it already when describing Fig. 3 in the results? If you want to keep it as a result, then simulations without preferential P regeneration might provide evidence for the causality you describe here. The same applies to p12,l10-11.

10. p12,l13-31: This section could be formulated a bit more concisely. Does the 1D model consider any N losses (at the bottom boundary, mimicked advection, ...)? If not I don't find it very surprising that newly fixed N accumulates in a 1D simulation - where else should it go - and would emphasize more the other aspects mentioned in this section.

11. p12,l25: misleading wording: "Our ... results ... show an accumulation of N .. which can only be explained by the new N input ...". This to me suggests that you tested different mechanisms for N accumulation. maybe better: "... accumulation resultion from N fixation".

12. p13,l27: misleading wording: "DIP is never exhausted ... because the lack of iron is hypothesized to prevent N2 fixation." maybe clearer: "DOP is never exhausted ... because the model implicitly assumes iron limitation to prevent N2 fixation."

**3   Technical corrections**

- title: results from a 1DV biogeochemical-physical coupled model: 1DV, not 1D? a typo?

- Fig. 4: the seasonal Phosphate gradient in the surface layers is difficult to distinguish with the current color scale. Can you find a better one?

- p7,l6-7: excessive dots within the units

- p8,l5: wording: the seasonal variation of the MLD ... It clearly indicates ...

- p8,l29: significant seasonal variation (no s)

- p13,l7: misplaced parentheses: (Olsen et al. 2016) instead of Olsen et al. (2016)

- p13,l8: leave out "the": To date, seasonal variations were ...

**3.1   reference formatting**

- italics for species names, i.e. Trichodesmium

- N 2 without space and with underscore

- CO2 with underscore

**3.2   additional spaces**

- p1,l4: after considered

- p2,l1: after NH4+

- p3,l14: after sampled

- p5,l8: after regard

**3.3   *Trichodesimum* should be *Trichodesmium**

- p8,l18 & 21

- p10,l23

---

## Referee Comment (RC2) · Anonymous Referee #2 · 4 Jun 2018

The study entitled "Diazotrophy as the main driver of planktonic production and bio-geochemical C, N, P cycles in theWestern Tropical South Pacific Ocean: results from a 1DV biogeochemical-physical coupled mode" by Gimenez et al. examines the role of diazotrophy in the western south Pacific ocean. The authors use a 1DV ecosystem model that is run in two different configurations at the same station that differ by the representation of diazotrophy. Their main conclusions are that nitrogen fixation sustains a significantly higher productivity, explains the consumption of DIP and induces

significant seasonal variations. Furthermore, they claim that a decoupling between the depth of the phosphacline and nitracline is necessary to induce nitrogen fixation. This manuscript addresses a very important scientific question: The potentially critical role of nitrogen fixation in Low Nutrient-Low Chlorophyll areas. It is relatively well written, clear and thus it deserves publication in Biogeosciences. However, some important issues have to be addressed before. In particular, some of the major conclusions proposed in that study are not sufficiently demonstrated and thus, require more analysis.

A strong point is made on the decoupling between the phosphacline and nitracline to explain nitrogen fixation. In fact, this is not clearly demonstrated in the study. It only shows that nitrogen fixation explains the lack of DIP accumulation at the surface. However, the impact of this decoupling on nitrogen fixation is not analysed. This would require to manipulate the depths of the nitracline and phosphacline independently and to study the consequences on nitrogen fixation. In the discussion section, the authors mention that they have done some sensitivity tests by altering the degradation rates of P-rich organic matters (DOP, POP), but the results of these tests are not shown. Yet, this would support their conclusions.

My second concern is on the model setup. First, they use the same physical conditions and state that the differences between WMA and WGY are only explained by the presence or lack of diazotrophy. That's quite a strong assumption that should be better discussed. Furthermore, the physical setup is not sufficiently described. For instance, they explain that they used the output of a WRF configuration to force their 1D ocean model without clearly describing the atmospheric model setup, the region that has been selected in this atmospheric model, how they have averaged (or not) the atmospheric forcing fields, . . . I don't expect a full dynamical validation of the physical state predicted by the physical 1D ocean model but some additional information are necessary. Finally, they have made some significant changes in their ocean biogeochemical models (two particles size classes, modified parameter values). They should explain why these changes have been made and how the parameter values have been

chosen (fine tuning, assimilation, basic assumptions, . . .).

My third concern is more subjective. I find the paper too descriptive and not quantitative enough. I would have liked to see a better quantification of the impact of nitrogen fixation on the system. Some budgets would have been interesting to present. For instance, how much of the PP is being sustained by nitrogen fixation directly (PP by UCYN and TRI) and indirectly (fresh input of N excreted by diazotrophs). How does N input from nitrogen fixation compare to export production? How sensitive are the model predictions to the parameter values, to UCYN and TRI descriptions, to DOP/POP dynamics? The latter question is partly related to my previous concerns.

To conclude on my mains concerns, I think that the authors should improve the analysis, perform some sensitivity tests, and better justify their choices and conclusions before this manuscript becomes suitable for publication.

Specific concerns:

P3, lines 24-27: This relates to one of my major concerns. The authors chose the same physical conditions to study two different sites. This is quite a strong assumption and this should be better justified. At present, I would consider that the study investigates the role of diazotrophy at a single location (which remains to be clearly stated, see below) rather than the differences between two sites.

P4, lines 20-21: I don't really understand what means TRI are equivalent to 100PHYL. I think that some more explanations in that paper would help the reader.

P5, lines 18-22: This is not clear enough. What is the model setup of WRF? Over which region and what time period have been averaged the atmospheric fields? How well does the physical model perform compared to actual in situ conditions?

P6, section 3.1.1: A major difference between the two simulations is the lack of DIP accumulation in the top 200m or so of the water column when diazotrophy is activated. I understand that in the top 50 or 70m of the water column where nitrogen fixation is

significant. However, below that depth range, DIP consumption is being increased in WMA without any significant N fixation. This should be explained.

P7, section 3.1.2: In the WGY setup, a very deep but intense DCM is predicted which is as strong as in the WMA configuration. Yet PP (and thus phytoplankton growth rates) is very very small in the DCM. How can you explain that, since grazing rates should be similar?

P8, sections 3.2.1 and 3.2.2: I understand that the lack of data prevents a detailed and complete validation of the seasonal dynamics predicted by the model. However, is it really impossible to do some basic validation using for instance satellite data for Chl, historical data for nutrients averaged over a regional box which would be valid since the model setup is not representative of a specific station but rather of a broad region.

P8, lines 16-18: I don't understand that statement. I don't understand why the vertical resolution of the fluxes is not the same as the vertical resolution of the state variables. It should be clarified.

P 9, section 4.1.1: the study suggests that DIP accumulation might by explained by the lack of nitrogen fixation. However, it does not explain why there are small rates of nitrogen fixation at WGY.

P10, lines 21-26: The authors performed an additional sensitivity run in which they suppress TRI but not UCYN. In that case, the DCM is shallower and the model skill is improved. However, does that lead to a complete exhaustion in DIP at the surface? In that case, the conclusion of the paper would be quite different since it would mean that low rates of N fixation as observed in WGY do not explain the DIP accumulation. Results from that sensitivity test should be presented in the manuscript.

P11, lines 10-21: the results of the sensitivity experiment mentioned in that paragraph should be included in the study as they directly support one the main conclusions, i.e. the decoupling between the phosphacline and the nitracline explains the high N fixation

rates at WMA.

P13, section 4.4: This section is a little bit too long and remains very descriptive.

---

## Author Comment (AC1) · 20 Jul 2018

**We greatly thank Referee #1 and #2 for their constructive comments and suggestions that we were pleased to answer during this revision process. All of them are detailed in the current .pdf file which contains :**

1) **The response to the Referee 1 comments**

2) **The response to the Referee 2 comments**

3) **The revised manuscript with modifications highlighted in blue**

4) **The supplementary material added to the revised manuscript**

**Response to Anonymous referee #1**

We thank Anonymous Referee #1 for the time and effort devoted to the review of the manuscript. Below, we reproduce the reviewer's comments and address their concerns point by point. The reviewer's comments are copied below in regular font with our responses in blue and the revised sections in the new manuscript version in red.

Just before answering, we would like to mention that during the revision process, we undertook some model technical check to answer a comment made by Referee #1. We thus noticed an error in the implementation of physical forcings, which was responsible for a two months shift between model results and data regarding the seasonal variations of the mixed-layer depth. We therefore ran again our two simulations with this error corrected. Corresponding results did not change at all our conclusion and were even improved compared to observations. This point is more detailed in the specific comment regarding Fig. 4, and the figures 3 and 4 of the manuscript will be consequently updated in the revised version.

**1 General comments**

The authors compare two 1D simulations only differing in the presence of diazotrophy to examine the role of N fixation for plankton production and biogoechemical cycles in observations in the Western Tropical South Pacific. While this aim of the study is given at in the introduction and the simulations presented are well suited to address this aim, in their interpretation of the results the authors claim to show the control of preferential P regeneration (a model assumption that is not tested) on N fixation. In my opinion, the simulations necessary to justify this latter claim are not provided. The results are very interesting and relevant to the current discussion on N fixation, and the underlying processes identified seem reasonable, but either the interpretation needs rephrasing or additional simulations need to be presented to acknowledge the causality implied by the study setup. Affected sections are identified in detail below. Furthermore, some validation of the simulated physics (MLD) on the seasonal scale would be needed to give confidence in the validity of the results.

1. Does the paper address relevant scientific questions within the scope of BG? yes

2.  Does the paper present novel concepts, ideas, tools, or data? yes

3. Are substantial conclusions reached? yes, but see below.

4. Are the scientific methods and assumptions valid and clearly outlined? partly. The design of the model study appears valid, but the way the interpretation of the results is phrased suggests a different cause and effect than the study design. Take, for example, the authors' claim in the abstract and the discussion that they "evidenced that the nitracline and phosphacline had to be respectively deeper and shallower than the Mixed-Layer Depth (MLD) .. [to create] ... favourable conditions for the development of diazotrophs" (p1,l8-11) and "concluded that a preferential regeneration of the detrital phosphorus (P) matter was necessary to obtain this gap between the nitracline and the phosphacline depths ..." (p1,l11-13). But neither the depth of the nutriclines nor the preferential P regeneration are manipulated in the simulations presented. Causality here goes the other way, in my opinion: the authors set up a system with preferential P regeneration and then show how N fixation creates different biogeochemical regimes.

The aim of our study was to investigate the role of nitrogen fixation in the biogeochemical contrast observed between the Western area (WMA) and the eastern area (WGY) of the WTSP by running two simulations, one representing WMA and the other WGY. For this, we first had to verify whether physical forcings were different in WMA and WGY and could explain part of this contrast between the two regions. When compared, atmospheric forcings and *in situ* mixed layer depths at WMA and WGY turned out to be very similar, which allowed us to use the same physical forcings for both simulations (we arbitrarily choose the forcings extracted at WMA). Furthermore, with the purpose of characterizing the role of diazotrophy in the contrast between WMA and WGY, the two simulations were designed so as to only differ by the presence or not of nitrogen fixers (present in WMA and absent in WGY). The two simulations simWMA and simWGY were run but they could not successfully represent WMA and WGY data unless modifying a feature of the biogeochemical model in order to reproduce the discrepancy between the nitracline and the phosphacline. In situ data indeed showed a nitracline deeper than the MLD (while the phosphacline is shallower) thereby allowing the input at the sea surface of P (but not N) from depth during winter mixing. To reproduce the nutriclines discrepancy, the only process-based lever in our model lied in the preferential regeneration of the P particulate matter, as this has already been measured in other regions. With this feature, the different biogeochemical fluxes and pools in WMA and WGY could have been well represented by the model, which allowed us to investigate the role of N2 fixation in the oligotrophy gradient observed in WTSP.

Nonetheless, we agree that more details and figures are necessary to illustrate the impact of this preferential P regeneration, and this point will be detailed in the specific comment "**causality between N fixation and preferential P regeneration".**

5. Are the results sufficient to support the interpretations and conclusions? no. The simulations required to show the control of P availability on N fixation as claimed (e.g., additional simulations without preferential P regeneration), are mentioned in the discussion as preliminary (p11,l34 - p12,l1), but are not shown. Either the interpretation needs to change from "control of P availability on N fixation" to "effects of N fixation on nutricline and seasonality", or additional simulations need to be provided.

We agree and decided for the sake of clarity to add the figures presenting the results from the simulation without preferential P regeneration, as detailed below in the specific comments "**causality between N fixation and preferential P regeneration".**

6. Is the description of experiments and calculations sufficiently complete and precise to allow their reproduction by fellow scientists (traceability of results)? no. The study refers to previous studies (Gimenez et al. 2016 - G2016, Alekseenko et al. 2014 - A2014) for all equations and parameter values, stating that "for all the non-diazotrophic features, TRI are parameterized as 100 PHYL cells ... and UCYN as PHYS." (p4, l20-21). Yet both references differ in some of the parameter values. Yet G2016 give specific mortality rates of UCYN as 1.16 $10^{-6}$ $s^{-1}$ compared to 1.16 $10^{-7}$ $s^{-1}$ for PHYS (A2014), and the max. growth rate of TRI as 2.08 $10^{-6}$ $s^{-1}$ compared to 2.3 $10^{-5}$ $s^{-1}$ for PHYL (A2014). This is confusing, although maybe not the fault of the authors: What is a typo, what is related to the conversion of values for TRI as 100 PHYL cells, which values are used in the present study?

Regarding the specific mortality rates, the value used for UCYN is the same as the one used for PHYS, namely 1.16 $10^{-6}$ $s^{-1}$. We confirm that there is a typo error in A2014 which also used the same mortality rate for PHYS (i.e. 1.16 $10^{-6}$ $s^{-1}$). Actually, the Eco3M-Med model firstly used and detailed in A2014 was then used by Guyennon et al. (2015)-G2015 and then used for our study. Obviously, during each of these works, additional reflexion and validation works led to some modifications and improvements regarding the model equations or parameters. For the sake of clarity, we propose here to provide our up-to-date complete set of parameter values in an additional table included in supplementary material (SM).

Regarding the TRI growth rate, we agree that the value reported in the manuscript is not well justified. In agreement with literature (e.g. Luo et al., 2012), we have considered in the model that a trichome of TRI was equivalent to 100 PHYL cells. This conversion factor between PHYL and TRI parameters was only applied for the "extensive parameters". Here extensive is used in the thermodynamic sense, and refers to the properties depending on the system size or the amount of material in the system. Since the specific growth rate is not an extensive property, this conversion factor was not applied to the specific growth rate of TRI, but we didn't use either the PHYL growth rate. Instead, we used a lower growth rate for TRI than for PHYL as suggested by experimental work: due to the filamentous shape of a trichome, a particular process for cell division is indeed observed in TRI. During their study on cell division in *Trichodesmium erythraeum* IMS101, Sandh et al. (2009) showed that "division never took place synchronously in the whole filament, but was restricted to small groups of cells that spread along the filaments" and that "the proportion of dividing cells, out of the total number of cells, varied from 5% to 20%". We therefore used the value of 0.17 d-1, which was averaged from values reported in the literature (Mulholland and Bernhardt, 2005, Hutchins et al., 2007).

We propose to clarify the part regarding the parameterization of diazotrophs (section 2.2.1) by adding :

"[..] For all the non-diazotrophic features and in agreement with literature (e.g. Luo et al., 2012), it has been considered that a Trichodesmium trichome was containing 100 PHYL cells and that a UCYN cell was equivalent to a PHYS cell. Yet, the conversion factor of 100 between TRI and PHYL was only applied for extensive parameters, i.e. those depending on biomass. Intensive parameters were set equal to those of PHYL, except for the specific growth rate which was instead averaged from literature since it has been experimentally demonstrated that it was lower than that of PHYL (Mulholland and Bernhardt, 2005, Hutchins et al., 2007). Parameter values, whether new or differing from those of Alekseenko et al. (2014), are given in SM: table 1."

7. Do the authors give proper credit to related work and clearly indicate their own new/original contribution? yes.

8. Does the title clearly reflect the contents of the paper? it reflects the study setup, but not the interpretation/conclusions drawn from it.

We propose to change the title by : "Diazotrophy as the main driver of the oligotrophic gradient in the Western Tropical South Pacific Ocean : results from a one-dimensional biogeochemical-physical coupled model".

9. Does the abstract provide a concise and complete summary? it needs rephrasing to fit the study design

The following sentence has been added to the abstract:

*"Since physical forcings in both regions were very similar, it was considered that the oligotrophic gradient observed in situ between WMA and WGY was not explained by differences in physical processes but rather by differences in biogeochemical processes. A one-dimensional physical-biogeochemical coupled model was thus used to [...]»*

10. Is the overall presentation well structured and clear? yes

11. Is the language fluent and precise? yes, only in few sections a bit redundant

12. Are mathematical formulae, symbols, abbreviations, and units correctly defined and used?

yes. some minor inconsistencies are pointed out below.

13. Should any parts of the paper (text, formulae, figures, tables) be clarified, reduced, combined, or eliminated? the abstract, discussion and conclusion should be brought in line with the simulations/results presented.

The different comments and questions raised during the revision process enabled us to improve the different sections of the paper in order to better present and highlight our main results and conclusion. All the modifications are presented in red in this current review response. They concern additional details on the nutriclines discrepancy due to a preferential regeneration of the P particulate matter : new figures have been added in the revised manuscript to better illustrate the model sensitivity (and especially on the N2 fixation rates) to this preferential P regeneration. The new figures and new sections associated are detailed in the specific concern "causality between N2 fixation and preferential P regeneration". We also improved the justification of our modeling strategy regarding the physical forcings implemented in the 1VD physical model by providing additional sections in the revised manuscript and figures in a supplementary material (details in the 1st specitic comment).

14. Are the number and quality of references appropriate? yes

15. Is the amount and quality of supplementary material appropriate? a supplement could clarify which parameter values were used (cf. no. 6 above)

We agree with Referee #1 and added a supplementary material to our article with the list of parameters used in our study and some additional figures.

**2 Specific comments**

**1. p3, l24-27:**

(a) you use two simulations identical in forcing and physics to simulate two locations about 40 deg. longitude apart with different biogeochemical characteristics. Is it justified to apply the same physics and forcing to both locations? If so, could you provide evidence for this, e.g. observations? It this choice compromising the fit between model and observations for station WGY?

As this point was also mentioned by Referee #2,  the same answer is given to both of them as follows :

We will first detail our choice of using the same physic forcing and then propose some new sentences which will be added to the revised manuscript in order to better justify our strategy.

 Our modeling strategy came from the observation that the WTSP was characterized by a significant biogeochemical gradient in terms of nutrient availability and planktonic production, which seemed to be directly related to the presence or not of nitrogen fixers inside this area (Moutin et al., 2018). In order to confirm or not this assumption through a modeling study, we designed two simulations only differing by the presence or not of diazotrophy. This was made possible only by the fact that, despite the large distance between WMA and WGY, the physical forcings were shown to be similar in the two  regions. The question of whether the atmospheric forcings in WMA and WGY were similar enough to consider that their impact on the water column dynamics was the same arose very early in our reflexion. In that purpose, at the early stage of this study, we compared the atmospheric forcings calculated at WMA and WGY by the atmospheric model WRF (see figure below).  This comparison shows that there is no significant difference between both forcings. To go further, we also compared two simulations only differing by the atmospheric forcing (respectively extracted in the two WMA and WGY regions) and did not observe differences nor in the water column dynamics, neither on biogeochemical cycles or on the trophic food web. Finally, in situ climatological data of MLD (Fig. 1) also indicate that there is no significant difference between the dynamics of these two regions despite their distance.  We thereby decided to use the atmospheric forcing from the WMA region for both regions and the simulated MLD predicted by our model fits well with values obtained with climatology (Fig. 1).  We acknowledge that this could have been further detailed in the submitted manuscript and propose to add the following text in the revised manuscript and the figure below in the supplementary material :

« The assumption made by using a unique set of atmospheric forcings for two regions significantly far away is first based on the *in situ* climatological data reported in Moutin et al. (2018). These authors showed that the vertical dynamics of the

water column, and especially the depths of the mixed layer were similar throughout the year in all the WTSP (see Figure SM1 in supplementary material). In addition, the atmospheric forcings calculated by the WRF atmospheric model at WMA and WGY were also very similar (see figure SM2 in supplementary material). Furthermore, we also compared two simulations ran with the respective atmospheric forcings calculated at WMA and WGY and did not observe any significant difference, nor in the water column dynamics, neither on the biogeochemical cycles. »

[Figure]

**Figure 1 (in SM)** Temporal dynamics of the *in situ* mixed layer depths estimated using a climatology (de Boyer Montégut et al., 2004) at WMA (green circles) and WGY (blue circles), and simulated by the model (green line)

[Figure]

**Figure 2 (in SM) Atmospheric forcings provided by the Weather Research Forecast model and extracted at the WMA (green) and WGY (blue) locations from September 2014 to September 2015**

(b) it confused me that the simulations were named after the different locations, e.g. simWGA, while applying the same physics/forcing. I was expecting different physics. In my words, I would say you used an idealized environment representative of station WMA to test the effect of N fixation, and got results in good agreement with the other location in the case of no N fixation. To me it would thus be more intuitive to call the simulations Nfixation and no-Nfixation or something similar.

We acknowledge that using the same atmospheric forcings for both simulations while referring to them with the name of two distinct regions could be quite confusing. Nonetheless, we finally found it better to use the simWMA/simWGY nomenclature as the aim of our study was to try to find explanations for the biogeochemical differences observed *in situ* between WMA and WGY stations. Since these stations have similar atmospheric forcings (we could have used as well a mean forcing between WMA and WGY but the results would have been identical), the comparison of both regions boils down to the comparison between simulations including or not diazotrophy. Moreover, using the same nomenclature for the model outputs and the observations makes their comparison easier and adds clarity to the text. This choice also emphasizes our conclusion which showed that, without diazotrophy, our model predicted biogeochemical features close to those observed in the WGY, and that the differences observed between the West and the East of the sampled transect during the cruise was not due to physical forcing but rather to the presence or not of N fixers.

**P4, l20-21.** How does your parameterisation of the different PFTs compare to other parameterisations of diazotrophy in the literature?

Regarding the model parametrisation, our strategy was to try as far as possible to keep the same parameter values than those used in the other Eco3M-Med model versions. By adding the N2 fixation process, we aimed to implement two new PFTs, a large one to represent *Trichodesmium* sp (TRI). and a small one to represent the UCYN (UCYN). We assumed that regarding their size, UCYN would be parameterized as a cell of small phytoplankton (PHYS). Indeed, ranging from 3 to 10 µm diameter (Zehr et al., 2011), the UCYN parameterization similar to PHYS was realistic as the PHYS compartment includes autotrophs <10 µm. The maximum specific growth rate used in our model for UCYN is 2.7 d-1, in good agreement with results from Agawing et al. (2007) who found a maximum specific growth rate of 2.0 d-1 for *Cyanothece* (UCYN-C) during a chemostat experiment.

The parameterization regarding TRI was already explained above in the 6[th] general comments.

Regarding the diazotrophy process, our model, based on the Rabouille et al. (2006) work on Trichodesmium, represents N2 fixation using the nitrogenase activity as a state variable. Rabouille et al. (2006) conducted a chemostat experiment in order to calibrate the set of parameters used to model the nitrogenase activity in Trichodesmium. We thus used the same formulation which calculates the nitrogenase activity with our own quota functions regulating the increase or decay of the nitrogenase activity (detailed in Gimenez et al., 2016 ). If we compare the N2 fixation rates modelled and observed during the chemostat experiment in Rabouille et al. (2006), we find that their maximum specific N2 fixation rates (11 mmolN.molC-1.h-1) are 4 times higher than the ones simulated in our model (2.8 mmolN.molC-1.h-1). As the maximum specific growth rate depends on the N and C intracellular contents and on the NO3 availability in the field, it is not surprising to observe lower rates in our more complex NPZD biogeochemical model than in the chemostat model used in Rabouille et al. (2006), which does not take into account for instance the resource competition between *Trichodesmium* and other organisms. Hence, these results show that our model calculates N2 fixation rates in the same order of magnitude than N2 fixation rates provided by other models or measured during the OUTPACE cruise (Fig. 3, c) in the manuscript).

In a work that could be similar to the one of Rabouille et al. (2006), Grimaud et al. (2013) combined a chemostat experiment and modelling to assess the influence of light-dark regime on the UCYN-B *Crocosphaera watsonii* metabolism. However, they did not represent the N2 fixation as a function of the nitrogenase activity like in Rabouille et et al. (2006), but through the nitrogenase pool, in terms of concentration, which depends on a fixed nitrogenase synthesis rate. We therefore kept the formulation used in Rabouille et al. (2006) which was adapted for smaller nitrogen fixers implemented in our model supposed to represent the unicellular nitrogen fixers (UCYN). There is thus no similar parameters in the literature to compare with the ones used in our model, however, if we compare the C specific N2 fixation rates observed in UCYN-B culture reported for instance by Fu et al. (2008), our simulated C specific N2 fixation rates by UCYN (1.4 nmolN.µmolC-1.h-1 in average) is in good agreement with the ones measured in their study ( from 1.2 to 3.2 nmolN.µmolC-1.h-1. For the same reasons as before, our model simulates N2 fixation rates close to the minimum range values observed in culture experiment, while the total N2 fixation rates simulated are consistent with the one measured during the OUTPACE cruise (Fig. 3, c) in the manuscript).

**P4, l29-p5,l12:** you modify the published biogeochemical model parameterization substantially. Why were those changes necessary? Did you perform any optimization or did you tune the model by hand? Was the model particularly sensitive to any of the parameters?

As this comment was also raised by Referee #2, we made the same response for both Referee #1 and Referee #2 :The biogeochemical model used in this study originates from a previous work as part of the VAHINE mesocosms experiment conducted in the Noumea Lagoon in New Caledonia, and published in Gimenez et al. (2016) - G2016. The model used in the G2016 study was run for 23 days and was not coupled to any physical model since it aimed at simulating a mesocosm experiment. Although G2016 and the present study use the same biogeochemical model and both aim to study the process of diazotrophy in the WTSP, the present work allowed to improve the first version of the biogeochemical model including nitrogen fixers and described in G2016. The entire year simulation run in this present study indeed revealed that some features of the model were not well represented, which led us to re-examine those features. The major problem we identified in the model results before its improvement was the replenishment in nitrate of the photic zone during the winter mixing which led to surface concentrations of nitrate largely higher than those measured *in situ*. This replenishment was due to a too shallow nitracline, itself controlled by the balance between uptake, sinking and mineralization of organic matter. In this regard, we studied the sensitivity of the model to those processes (without really performing an optimization) and found out that :

i) The sinking rates of organic matter was crucial in determining the depth of the nutriclines  To better represent the fate of the detrital matter as a function of its size and its source (i.e. large detrital particles come from large organisms and the small ones come from smaller organisms), the  detrital compartment has been splitted in two parts, each of them being associated with a different sinking rate.

The following paragraph has been added to the revised manuscript : «  Since the Gimenez et al. (2016) modelling study was focusing on a mesocosm experiment, the assessment of the model skills was incomplete.  With the new set of data

provided by the OUTPACE cruise, some features of the original model were improved and some new features were introduced to correct for the model major flaws or to add some realism in the model. To improve the representation of the nutricline depths which depend on the sinking of the detrital organic matter and its mineralization in the water column, we included two size classes of detrital matter associated with two different sinking rates, while the previous version (Gimenez et al., 2016) only included a single compartment of detrital material (see Table 1 in supplementary material). »

    ii)   DOP availability has a crucial role in this P-depleted area for autotrophs. Since DOP mineralization by heterotrophic bacteria was likely underestimated by the model because it does not explicitly represent P mineralization by ectoenzymes produced by bacteria, DOP availability for organisms has been artificially increased to take this phenomenon into account. In practice, half-saturation constants Ks for DOP uptake were divided by 10 for all autotrophs.

    iii)  As suggested by recent studies, we decided to apply the same Redfield ratio , i.e. 106:16:1 to the C:N:P ratios of the upper and lower ranges of intracellular quotas for all the PFTs, including bacteria and HNF for which the C:N:P ratios of 50:10:1 were used so far. This change also brought a more consistent stoichiometry in ciliates (CIL) which were predating so far on organisms with very different stoichiometries. We propose to insert the following paragraph to justify this choice :

«While several studies have shown that the intracellular C:N:P ratios in heterotrophic bacteria tend to be below Redfield values as they were enriched in N and P (Bratbak, 1985, Goldman and Dennett, 2000, Vrede et al., 2002 ), more recent studies suggest that these ratios could be higher than 50:10:1 and highly variables in response to physical, chemical and physiological conditions ( Cotner et al., 2010, Martiny et al., 2013 ; Zimmerman et al. 2014 ). This led us to replace the 50:10:1 ratios used so far in the model for bacteria and HNF by the Redfield 106:16:1 ratio as for the other PFTs represented in the model. It is reminded however that the PFT's stoichiometry is flexible in the model and that the Redfield ratios are only used to link together the limits of the ranges of C, N and P intracellular quotas, (i.e. $QC^{min} = 106\ QP^{min}$, $QC^{max} = 106\ QP^{max}$, see Table 1 in supplementary material) thereby allowing a large variety of possible C:N:P ratios in PFTs, and notably the 50:10:1 ratio in heterotrophic bacteria.»

**P5, l22; p8**,section 3.2.1.; Fig. 4: you mention winter and winter mixing. Is this appropriate for a tropical region at 17 deg S? As winter mixing I would understand much deeper and more rapid isolated mixing events as are characteristic for the temperate latitudes. Is the gradual deepening simulated for the MLD in Fig. 4 not mostly a result of increasing salinity in the dry season?

We thank Referee #1 for this relevant question regarding the cause of variations in the vertical density, and therefore in the vertical mixing observed during the austral winter (Moutin et al., 2018). Vertical mixing of the water column is induced by an increasing water density in the upper layer, which can either be due to a decreasing temperature or to an increasing salinity. A simple calculation of density variations using the equation of state of seawater, showed that temperature variations in the range of those measured in tropical regions (ΔT ~ 4,3 °C) result in density variations (Δ rho ~ 1,22), whereas no density variations were induced by the salinity variations typical in this region (Δ S ~ 0,004). This result shows therefore that, even in tropical waters, the vertical mixing is mostly due to a decreasing temperature in winter, though this decrease is not as important as in temperate latitudes. Moreover, around 22° N, Riser and Jonhson (2008) also observed « early winter mixing » which homogenized the upper water column at the HOT station, suggesting that a significant vertical mixing during the winter season can be observed even in tropical latitudes.

**p6,l18:** Could you mention here already that the shallower phosphacline than nitracline results from the model setup with preferential P regeneration? This aspect is not a result of the simulations presented here and would have made it significantly easier for me to understand this section. I acknowledge that there are different writing styles. In the case of this ms, readability of the results section would in my opinion be greatly improved if short explanations for the model data and the model-data discrepancies that follow directly from the model assumptions (i.e., the 1D model accumulating additional N due to missing advection; the shallower phosphacline compared to nitracline because of preferential P regeneration), were given already in the results section. The discussion section would then focus on putting the findings into context/analyzing more complex mechanisms.

We understand the point of Referee #1 although the Results sections aims to remain more descriptive than explanatory in our manuscript style. Nonetheless, we agree that the cause of the discrepancy between nitracline and phosphacline depths can be mentioned earlier than in the discussion, which is why we added the following sentence to introduce earlier the preferential P regeneration assumption.

« Note that to reproduce this discrepancy within the model, it has been necessary to introduce a preferential regeneration of the detrital matter in P compared to C and N in the biogeochemical model, a point thereafter detailed in Section 4.2.2. »

Fig. 2: could you add e.g. a small inset showing the location of this map on the globe?

Yes, the figure has been modified.

Fig. 3: you do not mention the identity of primary producers in the WMA simulation. With the confusing parameter value differences mentioned above I am not sure whether diazotrophs in these simulations only have net competitive advantages over the two other phyto types and would therefore be expected to dominate the WMA simulation?

We agree that giving the relative proportion of the different autotrophs to the total Chl a would have provided an evidence of the net competitive advantage of diazotrophs over the other non-diazotroph phytoplankton. Actually, in the photic layer, TRI represent 80% of the total Chl a, followed by PHYS (10%) and around 5 % each for UCYN and PHYL. The significant lower contribution of UCYN compared to TRI is due to the fact that UCYN are grazed by HNF and CIL while TRI has no predator in the model. In addition, the lower contribution of UCYN compared to PHYS seems counter-intuitive as we would think UCYN being advantaged by the N2 fixation compared to PHYS, but the energetic cost related to N2 fixation finally costs more to the cell than the advantage of fixing N2. These contributions were identical for PP. The following paragraph was added to the revised manuscript in order to give some details on the direct versus indirect impact of N2 fixation on other PFTs :

« In a previous study using nearly the same biogeochemical model including TRI and UCYN state variables in a one-dimensional configuration without physical coupling, Gimenez et al. (2016) highlighted the direct and indirect impact of the new N input provided by diazotrophs. By calculating the percentage of Diazotroph-Derived Nitrogen (DDN) in each model compartments, they followed the transfer of DDN throughout the entire trophic web as a function of time, and showed that after 25 days, 43% of the DDN fixed by diazotrophs were found in non-diazotroph organisms. These results clearly showed that N2 fixation had a significant indirect impact on the planktonic production by providing a new source of N for other organisms. DDN tracking inside the model compartments is associated with very high computational costs and could not be applied to the present study where simulations are run for several years (against 25 days for the previous study). However, it is worthwhile mentioning that the proportion of PFTs involved in total Chl a and PP was 80 % of TRI, 10 % of PHYS and 5% of PHYS and PHYL, suggesting that the impact of N2 fixation is rather direct (85% of PP is realized by diazotrophs) than indirect. In our model, diazotrophs have therefore net competitive advantage over the two other non-diazotrophic autotrophs. »

Moreover, instead of the fairly detailed description of the curves it might ease comprehension of the results to point out that in both simulations and observations the Chl a maximum is roughly located at the nutrient-limiting nutricline.

In response to this point, we will add the following sentence in the section 3.1.2 :

« In the model outputs, the location of the DCM is roughly located at the nutrient-limiting nutricline, i.e., at the phosphacline depth in WMA and simWMA, and at the nitracline depth in WGY and simWGY. »

Fig. 4: I guess the black line is the simulated MLD. You point out the connection between the MLD and the nutriclines on p11,ll-2. So how realistic is it compared to observations? Moutin et al. 2018 (BGD) present observation-based values from de Boyer Montegut et al. 2004 showing maximum MLD in August (here October) and high Chl from April to August (here October to April). What causes this discrepancy and how does it affect your results?

During the revision process and thanks to the reviewer's advice we spent a substantial time to figure out the time shift between the *in situ* MLD from the Deboyer Montegut climatology and the MLD predicted by the model. This allowed us to identify the source of the problem which was a technical error in the computer program used for the implementation of physical forcings in the model. The two simulations were then run again with the correction taken into account and, as expected, our main conclusions were not affected. The vertical profiles of Fig. 1 showed slight differences, but the corrected simulations fit better with observations than before (mostly the chlorophyll profiles in simWMA) and we are very grateful to the reviewer for his meticulous work. We revised the result section consequently by adding the new version of the figures, and removed the following sentence which mentioned the difference in the upper surface between the model and observations which is no more present with the corrected simulations :

« "[..] (around  16.0 nmolN.L$^{-1}$.d$^{-1}$ [..]

« [..] a winter mixing beginning at the end of  May leading to a maximum MLD of 70 m in  August."

«They clearly indicate a winter mixing beginning at the end of  May leading to a maximum MLD of 70 m in  August, followed by a longer stratified period from  November to  April,[..] »

« [..]During winter mixing, surface DIP$^{simW M A}$ increases from 0.6 to 2 nM, then remains quite stable until the end of  February before regularly decreasing until  June down to 0.6 nM. DIP$^{simW M A}$ then remains low during the stratified period until the next winter mixing in August. »

« The total $N_2$ fixation at the surface varies from a minimum mean value of 15 nmol.$L^{-1}.d^{-1}$ during the stratified period to a maximum mean value of 20 nmol.$L^{-1}.d^{-1}$ reached between  July and August, i.e. during the winter mixing . »

«The newly available DIP in the surface layer is immediately followed by an increase in the $N_2$ fixation rates in  June (Figure 3, c)), which then remain quite stable until  October before slightly decreasing until the next winter mixing [..]»

« [..] and a less intense and deeper signal around 70 m (which corresponds to the nitracline depthfrom  April to the end of  May). »

**causality between N fixation and preferential P regeneration:**

- p9,l29-p10,l2: "The ... role of DIP availability in controlling N2 fixation ... has been highlighted over the last decade ..., and the consistent results between the OUTPACE data and our model outputs, comparing simWGA and simWGY, reinforces this view of the biogeochemical functioning in the region." - You do not test the effects of P availability in the simulations presented, you assume high/sufficient P availability and manipulate N fixation. Your simulations thus show the connection the other way round: without N fixation you have P left over.

- p11,l10-11: as above: the simulations presented do not show that a shallower phosphacline than nitracline was needed to observe N fixation rates in agreement with observations. they show that assuming N fixation leads to this gap between phospha- and nitracline. Could it be that you still had the simulations without preferential P regeneration in mind, that are mentioned on p11,l34-p12,l1?
- p11,l23-25: I have to admit during the first read it took me until here to realize how the difference in phospha- and nutricline depths came about. The preferential P regeneration is a model assumption. Why not mention it already when describing Fig. 3 in the results? If you want to keep it as a result, then simulations without preferential P regeneration might provide evidence for the causality you describe here. The same applies to p12,l10- 11.

As this point on the causality between N fixation and preferential P regeneration was mentioned by the two referees, we made the same answer :

As mentioned above, we agree with Referee #1 regarding the lack of proof in the previous version of the manuscript to support the assumption of the P availability controlling N2 fixation and the preferential P regeneration leading to a shallower phosphacline than the nitracline. To remedy it, we decided to add the results of the simulation without the preferential P regeneration which were compared to the simWMA, in order to clearly show that, without that preferential regeneration, the system did not provide the sufficient conditions to support N2 fixation as observed in the field in the WMA region.

The following figures have been added to the revised manuscript in addition with the following paragraph, inserted at in Section 4.2.2 :

« To illustrate how the discrepancy between nitracline and phosphacline depths can be attributed to preferential P regeneration, DIN and DIP concentrations and N2 fixation rates in simWMA calculated with and without preferential P regeneration (i.e. preferential hydrolysis of P particulate matter) have been compared (Fig. 5). This figure shows the deepening of the phosphacline in the simulation without preferential P regeneration. More importantly, without preferential P regeneration, the phosphacline depth is deeper than the MLD at 70m, which prevents DIP input in the surface layer during winter mixing, and leads to a strong limitation of diazotrophs by P. This stronger P-limitation without preferential P regeneration can also be observed at the cellular scale by analyzing the intracellular P quota of N2 fixers (Figure 6). In the simulation without preferential P regeneration, the relative intracellular quota of P in TRI and UCYN

are in average respectively 8 and 16 times lower than with preferential P regeneration, as shown in Figure 6. As N2 fixation alleviates N limitation for diazotrophs, their growth is thus limited by P availability. The significant decrease in relative intracellular P quotas has therefore a significant impact on their growth, and consequently explain the lower N2 fixation rates shown in Figure 5, c). »

[Figure]

**Figure 3 (inserted as Figure 5 in the revised manuscript) Vertical profiles of (a) dissolved inorganic nitrogen (DIN), (b) dissolved inorganic phosphorus (DIP), (c) N2 fixation rates for the simulation with diazotrophy, as a proxy of the WMA region (simWMA with the preferential P regeneration (in green) and without the preferential P regeneration (in red). The horizontal black dashed line represents the depth of the maximum mixing layer calculated at 70 m depth.**

p12,l13-31: This section could be formulated a bit more concisely. Does the 1D model consider any N losses (at the bottom boundary, mimicked advection, ...)? If not I don't find it very surprising that newly fixed N accumulates in a 1D simulation - where else should it go - and would emphasize more the other aspects mentioned in this section.

As mentioned in this section, « [..] without any loss processes taken into account, the annual

N input by N2 fixation accumulates, as observed in simWMA. », there is no N losses in our model as it is totally conservative. Nonetheless, the main result pointed out in this section was not the accumulation of N, which is not surprising as mentioned by Referee #1 since N2 fixation brings new N in the system, but rather the location of this accumulation in the water column, namely around the main thermocline which seems on line with nitrate accumulation identified from N excess measurements (Fumenia et al., under rev.).

We propose to remove the following sentences which did not bring relevant information regarding the main result developed in this section, which is the location of the N accumulation in the water column : « The N* tracer is used to measure the N in excess with respect to the quantity expected from the thermocline N:P ratio (i.e., N:P of 16:1; Redfield et al. (1963); Takahashi et al. (1985); Anderson and Sarmiento (1994)), even if the relative contributions of the N-excess sources remain difficult to determine (Hansell et al., 2007). »

p12,l25: misleading wording: "Our ... results ... show an accumulation of N .. which can only be explained by the new N input ...". This to me suggests that you tested different mechanisms for N accumulation. maybe better: "... accumulation resultion from N fixation".

We agree with Referee #1 and propose the following rephrasing : « Our model […] in the 100-500 m layer which results from the new N input […].»

p13,l27: misleading wording: "DIP is never exhausted ... because the lack of iron is hypothesized to prevent N2 fixation."

maybe clearer: "DOP is never exhausted ... because the model implicitly assumes iron limitation to prevent N2 fixation."

We agree with Referee #1 for this rephrasing (assuming that DOP is a typo error instead of DIP), and the following sentence has been added to the revised manuscript.

« DIP is never exhausted [..] because the model implicitly assumes iron limitation to prevent N2 fixation ».

**Response to Anonymous referee #2**

We thank the anonymous Referee #2 for the time and effort devoted to the review of the manuscript. Below, we reproduce the reviewer's comments and address their concerns point by point. The reviewer's comments are copied below in regular font with our responses in blue and the revised sections in the new manuscript version in red.

Just before answering, we would like to mention that during the revision process, we undertook some model technical check to answer a comment made by Referee #1. We thus noticed an error in the implementation of the physical forcings, which was responsible for a two-month shift between model results and data regarding the seasonal variations of the mixed-layer depth. We therefore ran again our two simulations with this error corrected. Corresponding results did not change at all our conclusion, and were even improved compared to observations. This point is more detailed in the specific comment regarding Fig. 4, and the figures 3 and 4 of the manuscript will be consequently updated in the revised version.

The study entitled "Diazotrophy as the main driver of planktonic production and bio- geochemical C, N, P cycles in theWestern Tropical South Pacific Ocean: results from a 1DV biogeochemical-physical coupled mode" by Gimenez et al. examines the role of diazotrophy in the western south Pacific ocean. The authors use a 1DV ecosystem model that is run in two different configurations at the same station that differ by the representation of diazotrophy. Their main conclusions are that nitrogen fixation sustains a significantly higher productivity, explains the consumption of DIP and induces significant seasonal variations. Furthermore, they claim that a decoupling between the depth of the phosphacline and nitracline is necessary to induce nitrogen fixation. This manuscript addresses a very important scientific question: The potentially critical role of nitrogen fixation in Low Nutrient-Low Chlorophyll areas. It is relatively well written, clear and thus it deserves publication in Biogeosciences. However, some important issues have to be addressed before. In particular, some of the major conclusions proposed in that study are not sufficiently demonstrated and thus, require more analysis.

A strong point is made on the decoupling between the phosphacline and nitracline to explain nitrogen fixation. In fact, this is not clearly demonstrated in the study. It only shows that nitrogen fixation explains the lack of DIP accumulation at the surface. However, the impact of this decoupling on nitrogen fixation is not analysed. This would require to manipulate the depths of the nitracline and phosphacline independently and to study the consequences on nitrogen fixation. In the discussion section, the authors mention that they have done some sensitivity tests by altering the degradation rates of P-rich organic matters (DOP, POP), but the results of these tests are not shown. Yet, this would support their conclusions.

We fully agree with Referee #2 and decided to add the results of the simulation without preferential P regeneration and to provide more details on this important point. This point is detailed below in the 10[th] specific concern.

My second concern is on the model setup. First, they use the same physical conditions and state that the differences between WMA and WGY are only explained by the presence or lack of diazotrophy. That's quite a strong assumption that should be better discussed.

We agree with Referee #2 and developed this point below in the 1[st] specific concern.

Furthermore, the physical setup is not sufficiently described. For instance, they explain that they used the output of a WRF configuration to force their 1D ocean model without clearly describing the atmospheric model setup, the region that has been selected in this atmospheric model, how they have averaged (or not) the atmospheric forcing fields, . . . I don't expect a full dynamical validation of the physical state predicted by the physical 1D ocean model but some additional information are necessary.

We agree with Referee #2 and developed this point below in the 3[rd] specific concern.

Finally, they have made some significant changes in their ocean biogeo- chemical models (two particles size classes, modified parameter values). They should explain why these changes have been made and how the parameter values have been chosen (fine tuning, assimilation, basic assumptions, . . .).

As this comment was also raised by Referee #1, we made the same response for both Referee #1 and Referee #2 :The biogeochemical model used in this study originates from a previous work as part of the VAHINE mesocosms experiment conducted in the Noumea Lagoon in New Caledonia, and published in Gimenez et al. (2016) - G2016. The model used in the G2016 study was run for 23 days and was not coupled to any physical model since it aimed at simulating a mesocosm experiment. Although G2016 and the present study use the same biogeochemical model and both aim to study the process of diazotrophy in the WTSP, the present work allowed to improve the first version of the biogeochemical model including nitrogen fixers and described in G2016. The entire year simulation run in this present study indeed revealed that some features of the model were not well represented, which led us to re-examine those features. The major problem we identified in the model results before its improvement was the replenishment in nitrate of the photic zone during the winter mixing which led to surface concentrations of nitrate largely higher than those measured *in situ*. This replenishment was due to a too shallow nitracline, itself controlled by the balance between the sinking and the mineralization rates of organic matter. In this regard, we studied the sensitivity of the model to those processes (without really performing an optimization) and found out that :

i)    The sinking rates of organic matter was crucial in determining the depth of the nutriclines  To better represent the fate of the detrital matter as a function of its size and its source (i.e. large detrital particles come from large organisms and the small ones come from smaller organisms), the  detrital compartment has been splitted in two parts, each of them being associated with a different sinking rate.

The following paragraph has been added to the revised manuscript : «  Since the Gimenez et al. (2016) modelling study was focusing on a ponctual mesocosm experiment, the assessment of the model skills was incomplete.  With the new set of data provided by the OUTPACE cruise,   some features of the original model  were improved and some new features were introduced to correct the model major flaws or add some realism to the model. To improve the representation of the nutricline depths which depend on the sinking of the detrital organic matter and its mineralization in the water column, we included two size classes of detrital matter associated with two different sinking rates, while the previous version (Gimenez et al., 2016) only included a single compartment of detrital material (see Table 1 in supplementary material). »

ii)    DOP availability has a crucial role in this P-depleted area for autotrophs. Since DOP mineralization by heterotrophic bacteria was likely underestimated by the model because it does not explicitly represent P mineralization by ectoenzymes produced by bacteria, DOP availability for organisms has been artificially increased to take this phenomenon into account. In practice,  half-saturation constants Ks for  DOP uptake were divided by 10 for all autotrophs.

iii)   As suggested by recent studies, we decided to apply the  same Redfield ratio , i.e. 106:16:1 to the C:N:P ratios of the upper and lower ranges of intracellular quotas for all the  PFTs, including bacteria and HNF for which the C:N:P ratios of 50:10:1 were used so far.  This change also brought a more consistent stoichiometry in ciliates (CIL) which were predating so far on organisms with very different stoichiometries. We  propose to insert the following paragraph to justify this choice :

«While several studies have shown that the intracellular C:N:P  ratios in heterotrophic bacteria tend to be below Redfield values as they were enriched in N and P (Bratbak, 1985, Goldman and Dennett, 2000, Vrede et al., 2002 ), more recent studies suggest that these ratios could be higher than 50:10:1 and highly variables in response to physical, chemical and physiological conditions ( Cotner et al., 2010, Martiny et al., 2013 ; Zimmerman et al. 2014 ). This led us to replace the 50:10:1  ratios used so far in the model for bacteria and HNF by the Redfield 106:16:1 ratio as for the other PFTs represented in the model. It is reminded however that the PFT's stoichiometry is flexible in the model and that the Redfield ratios are only used to link together the limits of the ranges of C, N and P intracellular quotas, (i.e. $QC^{min} = 106 \ QP^{min}$, $QC^{max} = 106 \ QP^{max}$, see Table 1 in supplementary material) thereby allowing a large variety  of possible C:N:P ratios in PFTs, and notably the 50:10:1 ratio in heterotrophic bacteria.»

My third concern is more subjective. I find the paper too descriptive and not quantitative enough. I would have liked to see a better quantification of the impact of nitrogen fixation on the system. Some budgets would have been interesting to present. For instance, how much of the PP is being sustained by nitrogen fixation directly (PP by UCYN and TRI) and indirectly (fresh input of N excreted by diazotrophs). How does N input from nitrogen fixation compare to export production? How sensitive are the model predictions to the parameter values, to UCYN and TRI descriptions, to DOP/POP dynamics? The latter question is partly related to my previous concerns.

We thank Referee #2 for this concern, even though we find that the different points raised are quite out of the scope of this paper which did not focus on the fate of fixed N2 fixation on the system, but rather on the interactions between nutrients availability and N2 fixation at a seasonal scale. Budgets, details on how new N influences other compartments, or to what extent N2 fixation impacts export production, could be the material for another article of interest. We however added in the revised manuscript more details regarding the contribution of diazotrophs in PP rates and Chl a biomass to better highlight the clear advantage of TRI in simWMA compared to other autotrophs. We also mentioned that in a previous study, we investigated the transfer of the Diazotroph-Derived-Nitrogen in a similar but non-coupled biogeochemical model, and explain why it was not used in the present study considering the special issue time schedule.

« In a previous study using nearly the same biogeochemical model including TRI and UCYN state variables in a one-dimensional configuration without physical coupling, Gimenez et al. (2016) highlighted the direct and indirect impact of the new N input provided by diazotrophs.  By calculating the percentage of Diazotroph-Derived Nitrogen (DDN) in each model compartments, they followed the transfer of DDN throughout the entire trophic web as a function of time, and showed that after 25 days, 43% of the DDN fixed by diazotrophs were found in non-diazotroph organisms. These results clearly showed that N2 fixation had a significant indirect impact on the planktonic production by providing a new source of N for other organisms.  DDN tracking inside the model compartments is associated with very high computational costs and could not be applied to the present study where simulations are run for several years (against 25 days for the previous study). However, it is worthwhile mentioning that the proportion of PFTs involved in total Chl a and PP was 80 % of TRI, 10 % of PHYS and 5% of PHYS and PHYL, suggesting that the impact of N2 fixation is rather direct (85% of PP is realized by diazotrophs) than indirect. In our model, diazotrophs have therefore net competitive advantage over the two other non-diazotrophic autotrophs. »

To conclude on my mains concerns, I think that the authors should improve the analysis, perform some sensitivity tests : and better justify their choices and conclusions before this manuscript becomes suitable for publication.

More details and proofs regarding our assumptions and strategy were added in the revised manuscript. They concern additional details on the nutriclines discrepancy due to a preferential regeneration of the P particulate matter: new figures have been added in the revised manuscript to better illustrate the model sensitivity (and especially on the N2 fixation rates) to this preferential P regeneration. The new figures and new sections associated are detailed in the 10[th] specific concern. We also improve the justification of our modeling strategy regarding the physical forcings implemented in the 1VD physical model by providing additional sections in the revised manuscript and figures in a supplementary material (details in the 1[st] specific concern).

**Specific concerns:**

1- P3, lines 24-27: This relates to one of my major concerns. The authors chose the same physical conditions to study two different sites. This is quite a strong assumption and this should be better justified. At present, I would consider that the study investigates the role of diazotrophy at a single location (which remains to be clearly stated, see below) rather than the differences between two sites.

As this point was also mentioned by Referee #1, the same answer is given to both of them as follows :

We will first detail our choice of using the same physic forcing and then propose some new sentences which will be added to the revised manuscript in order to better justify our strategy.

Our modeling strategy came from the observation that the WTSP was characterized by a significant biogeochemical gradient in terms of nutrient availability and planktonic production, which seemed to be directly related to the presence or not of nitrogen fixers inside this area (Moutin et al., 2018). In order to confirm or not this assumption through a modeling study, we designed two simulations only differing by the presence or not of diazotrophy. This was made possible only by the fact that, despite the large distance between WMA and WGY, the physical forcings were shown to be similar in the two regions. The question of whether the atmospheric forcings in WMA and WGY were similar enough to consider that their impact on the water column dynamics was the same arose very early in our reflection. In that purpose, at the early stage of this study, we compared the atmospheric forcings calculated at WMA and WGY by the atmospheric model WRF (see figure below). This comparison shows that there is no significant difference between both forcings. To go further, we also compared two simulations only differing by the atmospheric forcing (respectively using the to WMA and WGY forcings) and did not observe differences nor in the water column dynamics, neither on biogeochemical cycles or on the trophic web. Finally, in situ climatological data of MLD (Fig. 1) also indicate that there is not significant difference between the dynamics of these two regions despite their distance. We thereby decided to use the atmospheric forcing from the WMA region for both regions and the simulated MLD predicted by our model fits well with values obtained with climatology (Fig. 1). We acknowledge that this could have been further detailed in the submitted manuscript and we propose to add the following text in the revised manuscript and the figure below in the supplement material :

« The assumption made by using a unique set of atmospheric forcings for two regions significantly far away is first based on the in situ climatological data reported in Moutin et al. (2018). These authors indeed showed that the vertical dynamics of the water column, and especially the depths of the mixed layer were similar throughout the year in all the WTSP (see Figure SM1 in supplementary material). In addition, the atmospheric forcings calculated by the WRF atmospheric model at WMA and WGY were also very similar (see Figure SM2 in supplementary material). Furthermore, we also compared two simulations ran with the respective atmospheric forcings calculated at WMA and WGY and did not observe any significant difference, nor in the water column dynamics, neither on biogeochemical cycles. »

[Figure]

**Figure 1 (in SM)** Temporal dynamics of the *in situ* mixed layer depths estimated using a climatology (de Boyer Montégut et al., 2004) at WMA (green circles) and WGY (blue circles), and simulated by the model (green line)

[Figure]

**Figure 2 (in SM)** **Atmospheric forcings provided by the Weather Research Forecast model and extracted at the WMA (green) and WGY (blue) locations from September 2014 to September 2015**

2 - P4, lines 20-21: I don't really understand what means TRI are equivalent to 100PHYL. I think that some more explanations in that paper would help the reader.

The following sentence has been modified in the revised manuscript to ease the comprehension:

"[..] For all the non-diazotrophic features and in agreement with literature (e.g. Luo et al., 2012), it has been considered that a Trichodesmium trichome was containing 100 PHYL cells and that a UCYN cell was equivalent to a PHYS cell. Yet, the conversion factor of 100 between TRI and PHYL was only applied for extensive parameters, i.e. those depending on biomass. Intensive parameters were set equal to those of PHYL, except for the specific growth rate which was instead averaged from literature since it has been experimentally demonstrated that it was lower than that of PHYL (Mulholland and Bernhardt, 2005, Hutchins et al., 2007). Parameter values, whether new or differing from those of Alekseenko et al. (2014), are given in SM: table 1."

3 - P5, lines 18-22: This is not clear enough. What is the model setup of WRF? Over which region and what time period have been averaged the atmospheric fields? How well does the physical model perform compared to actual in situ conditions?

As mentioned in the manuscript in section 2.3, the atmospheric forcings were provided by a run of the WRF model starting at the end of the winter mixing period preceding the OUTPACE cruise and lasting one year (i.e. Sept. 2014 – Sept. 2015)

. Concerning the location, the atmospheric forcings were extracted at the precise location of the long duration station sampled in the WMA region during the cruise, i.e. the LD A station, located at 19,2°S 164,7°E. There was thus no space or time average for the atmospheric forcings. Boundary conditions for the WRF model are provided by the American Global Forecast System (GFS) model (National Center for Environmental Prediction/National Center Environmental Prediction -NCAR / NCEP) analyzes. These analyzes correspond to a correction of the forecast using a larger number of observations during the data assimilation cycle. The WRF model is forced every 6 hours by analyzes during the processing.

The following paragraph has been added to the revised manuscript:

« [..15 km] and a time step of 6 hours. Boundary conditions for the WRF model are provided by the American Global Forecast System (GFS) model (National Center for Environmental Prediction/National Center Environmental Prediction - NCAR / NCEP) analyzes. These analyzes correspond to a correction of the forecast using a larger number of observations during the data assimilation cycle. The WRF model is forced every 6 hours by analyzes during the processing. »

Regarding the validation of the physical model, seasonal in situ data over the simulated period (Sept. 2014 – Sept. 2015) were not available to compare with the outputs of our physical model. However, we used climatology data to assess this model and found a good consistency between simulated  and  field-derived MLD in the WTSP (see  Moutin et al. (2018)). The figures below represent the monthly-averaged surface temperature and density and the depth of the mixed layer provided by the physical model, compared to temperature and density data extracted from the WOA13 climatology (2005-2012)  and MLD data from the de Boyer Montégut et al. (2004) climatology. We propose to join this figure to support the validation of our physical model and the following sentence has been added in Section 2.3 :

« The comparison between some physical outputs (i.e. surface temperature, surface density, mixed layer depth) and climatological *in situ* observations allowed to ensure that the one-dimensional physical model was relevant to address our scientific question  (see Figure SM3 in supplementary material) .»

[Figure]

[Figure]

[Figure]

**Figure 3 in SM** **Evolution of monthly averaged (a) sea surface temperature (SST), (b) surface density and (c) mixed layer depths (MLD) from September 2014 to August 2015 predicted by the model (green line) and calculated with climatologies (WOA13 for SST and Surface density, and de Boyer Montegut et al., 2004 for MLD)**

4 - P6, section 3.1.1: A major difference between the two simulations is the lack of DIP accumulation in the top 200m or so of the water column when diazotrophy is activated. I understand that in the top 50 or 70m of the water column where

nitrogen fixation is significant. However, below that depth range, DIP consumption is being increased in WMA without any significant N fixation. This should be explained.

Section 3.1.1 describes Fig. 3 a)-b)-c) in order to correlate the vertical profiles of nutrients and that of N2 fixation. As mentioned by Referee #2, the main difference between simWMA and simWGY remains in the upper 0 to 70m layer, where N2 fixation is significant and DIP depleted in simWMA while a DIP accumulation is observed in simWGY without N2 fixation.

However, we are not sure to fully understand the second point mentioned by Referee #2 regarding DIP consumption. We can indeed notice in Fig. 3 b) that the phosphacline in simWMA starts at 70 m where the DIP concentration starts to increase gradually until 300 m depth. This means that below 70 m (which corresponds to the bottom of the photic zone), where there is indeed no more N2 fixation (Fig. 3 c)), DIP is less consumed by organisms and accumulates, and this is why we observe the beginning of the phosphacline at this depth. In simWGY, without N2 fixation, DIP concentrations are higher in the upper surface layer than in simWMA because the system is N-limited, thus preventing P consumption. This is why the organisms develop deeper, around the depth of the nitracline in simWGY.

5 - P7, section 3.1.2: In the WGY setup, a very deep but intense DCM is predicted which is as strong as in the WMA configuration. Yet PP (and thus phytoplankton growth rates) is very very small in the DCM. How can you explain that, since grazing rates should be similar?

As mentioned by Referee #2, grazing kinetics are the same in simWMA and simWGY but the effective grazing rates are not necessarily the same since they are also function of predator and preys concentrations. Moreover, though nitrogen fixation only occurs at simWMA, the DCM intensity is the same in both regions. The vertical profiles of PP and Chl a show that PP values are not proportional to the DCM concentration, and that the PP and Chla maxima do not necessarily match (it matches for WGY but not for WMA). For the same DCM intensity, PP values are indeed much lower in simWGY than in simWMA. A possible explanation for this lies in the fact that the composition of both DCM are quite different: the DCM in simWMA corresponds to high C biomass (i.e. the Chl:C ratios of autotrophs is quite low), while the DCM in simWGY is composed by less autotrophs which have higher Chl:C ratios (since they develop deeper where light is lower, they have to generate more Chl a pigments to increase their C fixation rate).

6 - P8, sections 3.2.1 and 3.2.2: I understand that the lack of data prevents a detailed and complete validation of the seasonal dynamics predicted by the model. However, is it really impossible to do some basic validation using for instance satellite data for Chl, historical data for nutrients averaged over a regional box which would be valid since the model setup is not representative of a specific station but rather of a broad region.

We agree with Referee #2 that such a validation would have potentially brought more confidence in the seasonal variations calculated by our model. However, in this oligotrophic to ultra-oligotrophic area, Chl a variations at the surface are very low and rarely observed in data, compared to other systems where large Chl a variations can be observed at the ocean surface. Moreover, the variations observed in the DIP concentrations in the model (Fig. 4, b)) remain in a range values below the quantification limit of the classical DIP measurements, meaning that this slight variations cannot commonly be observed *in situ*. Moreover, the complete list of previous cruises in this area is detailed in Fumenia et al. uner rev. showing that we are far to get an annual survey in this under sampled area. However, despite of the absence of possible validation, we considered that the results provided by the model were plausible and interesting enough to be analyzed and published in the present paper.

7 - P8, lines 16-18: I don't understand that statement. I don't understand why the vertical resolution of the fluxes is not the same as the vertical resolution of the state variables. It should be clarified.

The standard model outputs of the biogeochemical model consist in the vertical profiles of the different concentrations (pools) of all the state variables represented in the model, as a function of time. The dynamics of each state variable is calculated in the model thanks to a combination of several fluxes, which represent the different biological processes controlling the dynamics (growth, grazing, mortality, nutrients uptakes, exudation, etc..). However, the systematic storage of the numerical values of all these fluxes is impossible accounting for the computer memory this would require. Instead, we must select some fluxes and depths of interest to be saved. Since no significant N2 fixation or PP rates were measured below 100 m, we did not save the numerical values of these fluxes below 100 m.

The following sentence has been added in the revised manuscript at the end of section 3.2.1 :

«[..] Accounting for the huge computer memory this would require, the values of the different biogeochemical fluxes calculated by the model are not systematically saved. As a result, numerical values of fluxes are saved at a lower vertical resolution than concentrations (pools). For this reason why [..] »

8 - P 9, section 4.1.1: the study suggests that DIP accumulation might by explained by the lack of nitrogen fixation. However, it does not explain why there are small rates of nitrogen fixation at WGY.

We understand the point raised here by Referee #2 and acknowledge that the reason of the very low rates of N2 fixations observed in WGY are not detailed in this section. While this point is related to one of our main assumption, the scope of

this paper was not to provide explanations of the N2 fixation gradient observed during the cruise, as it was already described in Bonnet et al. (2017), Moutin et al. (2018) and Guieu et al. (2018). In order to bring some additional explanation to the reader, the following sentences have been added to the revised manuscript :

« Because of the high Fe requirement of diazotrophs (Paerl et al., 1987; Rueter et al., 1990), the low Fe availability in WGY is assumed to prevent or significantly limit N2 fixation in the South Pacific gyre (Moutin et al., 2008; Guieu et al., 2018, Moutin et al., 2018). By contrast, the high Fe availability in WMA is assumed to favor the growth of nitrogen fixers. Guieu et al. (2018) indeed measured high DFe concentrations in the photic layer in WMA provided by abnormally shallow hydrothermal sources (around 500 m deep) in the WTSP. Due to very low N2 fixation measured in WGY (Bonnet et al., 2018), autotrophs organisms are N-limited, leading to a lower PP than in WMA, which results in a higher DIP accumulation in the photic layer since DIP is less consumed by organisms. »

9 - P10, lines 21-26: The authors performed an additional sensitivity run in which they suppress TRI but not UCYN. In that case, the DCM is shallower and the model skill is improved. However, does that lead to a complete exhaustion in DIP at the surface? In that case, the conclusion of the paper would be quite different since it would mean that low rates of N fixation as observed in WGY do not explain the DIP accumulation. Results from that sensitivity test should be presented in the manuscript.

We thank Referee #2 for this interesting comment. This simulation led to the following conclusions:

-N2 fixation was significantly lower than the one in simWMA, but still 4 times higher than the rates calculated in simWGY.

- DIP concentrations were lower than in simWGY but still more than 10 times higher than in simWMA. In other words, this simulation did not lead to a complete exhaustion of available P in the upper surface layer.

We realize that the way this simulation is presented at the end of Section 4.1.2 might suggest that it is a better proxy of the ecosystem sampled in WGY than simWGY, but this is not the case. It is rather an intermediate system, with higher N2 fixation rates than those measured in WGY. We finally decided to rephrase the paragraph of interest without presenting the figures, as these figures would not contribute to illustrate one of our major result but concerns only a simulation used to argue the deeper DCM predicted in simWGY compared to the one measured in WGY. We reckon that presenting results from an additional simulation would confuse the guideline of the article.

« The results of this intermediate simulation (not shown) indicate low surface PP rates and POC concentrations, in agreement with those measured in WGY. Moreover, dissolved P is still available in the photic zone (though at concentrations lower than for simWGY) even if the calculated N2 fixation rates were slightly higher than the measured ones. In addition, the DCM (located around 150 m) and the nutriclines were shallower than in simWGY (i.e. without any diazotrophs). This simulation can thus be considered like an intermediate system between simWMA and simWGY, and confirms the close link between N2 fixation fluxes and P availability. »

10 - P11, lines 10-21: the results of the sensitivity experiment mentioned in that paragraph should be included in the study as they directly support one the main conclusions, i.e. the decoupling between the phosphacline and the nitracline explains the high N fixation rates at WMA.

As this point on the causality between N fixation and preferential P regeneration was mentioned by the two referees, we made the same answer :

As mentioned above, we agree with Referee #1 regarding the lack of proof in the previous version of the manuscript to support the assumption of the P availability controlling N2 fixation and the preferential P regeneration leading to a shallower phosphacline than the nitracline. To remedy it, we decided to add the results of the simulation without the preferential P regeneration which were compared to the simWMA, in order to clearly show that, without that preferential regeneration, the system did not provide the sufficient conditions to support N2 fixation as observed in the field in the WMA region.

The following figures have been added to the revised manuscript in addition with the following paragraph, inserted at the end of Section 4.2.2 :

« To illustrate how the discrepancy between nitracline and phosphacline depths can be attributed to preferential P regeneration, DIN and DIP concentrations and N2 fixation rates in simWMA calculated with and without preferential P regeneration (i.e. preferential hydrolysis of P particulate matter) have been compared (Fig. 5). This figure shows the deepening of the phosphacline in the simulation without preferential P regeneration. More importantly, without preferential P regeneration, the phosphacline depth is deeper than the MLD at 70m, which prevents DIP input in the surface layer during winter mixing, and leads to a strong limitation of diazotrophs by P. This stronger P-limitation without preferential P regeneration can also be observed at the cellular scale by analyzing the intracellular P quota of N2 fixers. In the simulation without preferential P regeneration, the relative intracellular quota of P in TRI and UCYN are in average respectively 8 and 16 times lower than with preferential P regeneration. As N2 fixation alleviates N limitation for diazotrophs, their growth is thus limited by P availability. The significant decrease in relative intracellular P quotas has

therefore a significant impact on their growth, and consequently explain the lower N2 fixation rates showed in Figure 5, c). »

[Figure]

**Figure 4 (inserted as Figure 5 in the revised manuscript) Vertical profiles of (a) dissolved inorganic nitrogen (DIN), (b) dissolved inorganic phosphorus (DIP), (c) N2 fixation rates for the simulation with diazotrophy, as a proxy of the WMA region (simWMA with the preferential P regeneration (in green) and without the preferential P regeneration (in red). The horizontal black dashed line represents the depth of the maximum mixing layer calculated at 70 m depth.**

11 - P13, section 4.4: This section is a little bit too long and remains very descriptive.

We acknowledge that this section is quite long and it has been shortened and rewritten as follows in the revised manuscript:

[revised manuscript text omitted]

**Supplementary material**

| Symbol | Definition | Units | Value HNF | Value BAC | Value PHYS | Value UCYN | Value PHYL | Value TRI |
|---|---|---|---|---|---|---|---|---|
| | | | | | | | | |
| | | *DOP assimilation* | | | | | | |
| $K_{LDOP}$ | Half-saturation constant for LDOP | mol.L$^{-1}$ | - | $6.62\ 10^{-7}$ | $6.57\ 10^{-7}$ | $6.57\ 10^{-7}$ | $5.66\ 10^{-6}$ | $5.66\ 10^{-6}$ |
| | | *Intracelullar contents* | | | | | | |
| $Q_P^{min}$ | minimum phosphate content | mol.cell$^{-1}$ | $1.27\ 10^{-12}$ | $1.15\ 10^{-15}$ | - | - | - | - |
| $Q_P^{max}$ | maximum phosphate content | mol.cell$^{-1}$ | $3\ Q_P^{min}$ | $3\ Q_P^{min}$ | - | - | - | - |
| $Q_N^{min}$ | minimum nitrogen content | mol.cell$^{-1}$ | $16\ Q_P^{min}$ | $16\ Q_P^{min}$ | - | - | - | - |
| $Q_N^{max}$ | maximum nitrogen content | mol.cell$^{-1}$ | $3\ Q_N^{min}$ | $3\ Q_N^{min}$ | - | - | - | - |
| $Q_C^{min}$ | minimum carbon content | mol.cell$^{-1}$ | $106\ Q_P^{min}$ | $106\ Q_P^{min}$ | - | - | - | - |
| $Q_C^{max}$ | maximum carbon content | mol.cell$^{-1}$ | $3\ Q_C^{min}$ | $3\ Q_C^{min}$ | - | - | | |
| | | *Nutrients assimilation* | | | | | | |
| $V_{\text{NO3}}^{max}$ | Maximum uptake rate for NO$_3$ | mol.cell$^{-1}$.s$^{-1}$ | $\mu \cdot Q_N^{max}$ | $\mu \cdot Q_N^{max}$ | - | - | - | - |
| $V_{\text{NH3}}^{max}$ | Maximum uptake rate for NH$_4$ | mol.cell$^{-1}$.s$^{-1}$ | $\mu \cdot Q_N^{max}$ | $\mu \cdot Q_N^{max}$ | - | - | - | - |
| $V_{\text{PO4}}^{max}$ | Maximum uptake rate for PO$_4$ | mol.cell$^{-1}$.s$^{-1}$ | $\mu \cdot Q_P^{max}$ | $\mu \cdot Q_P^{max}$ | - | - | - | - |
| $V_{\text{DON}}^{max}$ | Maximum uptake rate for DON | mol.cell$^{-1}$.s$^{-1}$ | $\mu \cdot Q_N^{max}$ | $\mu \cdot Q_N^{max}$ | - | - | - | - |
| $V_{\text{DOP}}^{max}$ | Maximum uptake rate for DOP | mol.cell$^{-1}$.s$^{-1}$ | $\mu \cdot Q_N^{max}$ | $\mu \cdot Q_P^{max}$ | - | - | - | - |

| Symbol | Definition | Units | DETS-C | DETL-C | DETS-N | DETL-N | DETS-P | DETL-P |
|---|---|---|---|---|---|---|---|---|
| | | *Particulate matter hydrolysis and sink* | | | | | | |
| $\omega$ | sinking rate | m.d$^{-1}$ | 1.0 | 25.0 | 1.0 | 25.0 | 1.0 | 25.0 |
| $\text{TT}_{DET_P}$ | Turnover time for DET-P | d$^{-1}$ | | | | | 0.5 | 0.5 |

**Table 1.** Model parameters which differ from Alekseenko et al. (2014) mentioned in Section **2.2.2**, with $\mu$ = maximum growth rate

| Symbol | Definition | Value TRI | Value UCYN | Units - |
|---|---|---|---|---|
| | | *Growth and Intracelullar contents* | | |
| $\mu_{max}$ | maximum growth rate | $2.08\ 10^{-6}$ | $3.2\ 10^{-5}$ | $s^{-1}$ |
| $k_m$ | specific natural mortality rate | $1.16\ 10^{-6}$ | $1.16\ 10^{-6}$ | $s^{-1}$ |
| $Q_C^{min}$ | minimum cell quota of C | $2.28\ 10^{-10}$ | $6.84\ 10^{-15}$ | $molC.Cell^{-1}$ |
| $Q_C^{max}$ | maximum cell quota of C | $6.84\ 10^{-15}$ | $2.05\ 10^{-14}$ | $molC.Cell^{-1}$ |
| $Q_N^{min}$ | minimum cell quota of N | $3.44\ 10^{-11}$ | $1.03\ 10^{-15}$ | $molN.Cell^{-1}$ |
| $Q_N^{max}$ | maximum cell quota of N | $1.03\ 10^{-10}$ | $3.09\ 10^{-15}$ | $molN.Cell^{-1}$ |
| $Q_P^{min}$ | minimum cell quota of P | $3.44\ 10^{-11}$ | $1.03\ 10^{-15}$ | $molN.Cell^{-1}$ |
| $Q_P^{max}$ | maximum cell quota of P | $1.03\ 10^{-10}$ | $3.09\ 10^{-15}$ | $molN.Cell^{-1}$ |
| $Q_{CN}^{min}$ | minimum cell C:N ratio | $5.0$ | $5.0$ | $molC.molN^{-1}$ |
| $Q_{CN}^{max}$ | maximum cell C:N ratio | $19.8$ | $19.8$ | $molC.molN^{-1}$ |
| $Q_{CP}^{min}$ | minimum cell C:P ratio | $35.33$ | $35.33$ | $molC.molP^{-1}$ |
| $Q_{CP}^{max}$ | maximum cell C:P ratio | $318.0$ | $318.0$ | $molC.molP^{-1}$ |
| | | *Nutrients assimilation* | | |
| $K_{NO3-}$ | Half-saturation constant for $NO3-$ | $1.85\ 10^{-6}$ | $7.6\ 10^{-6}$ | $mol.L^{-1}$ |
| $V_{NO3-}^{max}$ | Maximum uptake rate for $NO3-$ | $3.16\ 10^{-15}$ | $9.91\ 10^{-20}$ | $mol.cell^{-1}.s^{-1}$ |
| $K_{NH4+}$ | Half-saturation constant for $NH4+$ | $7.0\ 10^{-6}$ | $1.69\ 10^{-6}$ | $mol.L^{-1}$ |
| $V_{NH4+}^{max}$ | Maximum uptake rate for $NH4+$ | $3.16\ 10^{-15}$ | $9.91\ 10^{-20}$ | $mol.cell^{-1}.s^{-1}$ |
| $K_{PO_4^{3-}}$ | Half-saturation constant for $PO_4^{3-}$ | $1.4\ 10^{-6}$ | $2.62\ 10^{-7}$ | $mol.L^{-1}$ |
| $V_{PO_4^{3-}}^{max}$ | Maximum uptake rate for $PO_4^{3-}$ | $1.98\ 10^{-16}$ | $6.19\ 10^{-21}$ | $mol.cell^{-1}.s^{-1}$ |
| $K_{DON}$ | Half-saturation constant for $DON$ | $4.32\ 10^{-5}$ | $1.05\ 10^{-5}$ | $mol.L^{-1}$ |
| $V_{DON}^{max}$ | Maximum uptake rate for $DON$ | $3.16\ 10^{-15}$ | $9.91\ 10^{-20}$ | $mol.cell^{-1}.s^{-1}$ |
| $K_{DOP}$ | Half-saturation constant for $DOP$ | $3.4\ 10^{-6}$ | $6.57\ 10^{-7}$ | $mol.L^{-1}$ |
| $V_{DOP}^{max}$ | Maximum uptake rate for $DOP$ | $3.16\ 10^{-15}$ | $6.19\ 10^{-21}$ | $mol.cell^{-1}.s^{-1}$ |
| | | *Diazotrophy process* | | |
| $Nase_{prod}^{max}$ | Maximum rate of increase of nitrogenase activity | $1.17\ 10^{-21}$ | $3.51\ 10^{-26}$ | $mol.cell^{-1}.s^{-2}$ |
| $Nase_{decr}^{max}$ | Maximum rate of decay of nitrogenase activity | $9.36\ 10^{-22}$ | $2.83\ 10^{-26}$ | $mol.cell^{-1}.s^{-2}$ |
| $K_{Nase}$ | Coefficient of nitrogenase degradation | $9.44\ 10^{-16}$ | $1.92\ 10^{-20}$ | $mol.cell^{-1}.s^{-1}$ |
| $COST_{DIAZO}$ | Respiration cost for nitrogen fixation | $1.5$ | $1.5$ | $mol.mol^{-1}$ |
| $EXUD_{DON}$ | Exudation part of $N_2$ fixed towards $DON$ | $0.5$ | $0.5$ | |
| $EXUD_{NH_4}$ | Exudation part of $N_2$ fixed towards $NH_4$ | $0.5$ | $0.5$ | |

**Table 2.** Model Parameters relative to diazotroph organisms TRI and UCYN

[Figure]

**Figure SM 1** Temporal dynamics of the *in situ* mixed layer depths estimated using a climatology (de Boyer Montégut et al., 2004) at WMA (green circles) and WGY (blue circles), and simulated by the model (green line)

[Figure]

**Figure SM 2** Atmospheric forcings provided by the Weather Research Forecast model and extracted at the WMA (green) and WGY (blue) locations from September 2014 to September 2015

[Figure]

[Figure]

[Figure]

**Figure SM 3** Evolution of monthly averaged (a) sea surface temperature (SST), (b) surface density and (c) mixed layer depths (MLD) from September 2014 to August 2015 predicted by the model (green line) and calculated with climatologies (WOA13 for SST and Surface density, and de Boyer Montegut et al., 2004 for MLD)

---

## Author Comment (AC2) · 20 Jul 2018

The comment was uploaded in the form of a supplement:
https://www.biogeosciences-discuss.net/bg-2018-162/bg-2018-162-AC2-supplement.pdf

---

## Author Comment (AC3) · 20 Jul 2018

**Response to E. Hrustic**

We sincerely thank E. Hrustic for the time and effort devoted to the review of the manuscript. Below, we reproduce the E. Hrustic's comments and address their concerns point by point. The reviewer's comments are copied below in regular font with our responses in blue and the revised sections in the new manuscript version in red.

Dear authors,

I have read this manuscript with a great interest since the topic is higly important for the understanding of the complexity of the primary productivity controlling factors and the role of N 2 fixation in nutrient limited oceans.

We would like to greatly thank E. Hrustic for these constructive comments and advices which led us to significantly improve our manuscript. The three hereafter points have been answered in details below, and most of the technical details listed below were took into account in the revised version

**General comments:**
I find that the manuscript brings interesting combination of field research with modelling. Also, the exploration of nitracline and phosphacline depths variations in oligotrophic and ultra-oligotrophic regions of WTSP, including another analyses, enables a discussion of a high quality. Presented results are clear inspite of the complexity of the conducted research. I find the manuscript excellent, according to my skills to judge an overall scientific contribution. At the same time, I see some possibilites for improvements in Introduction, Methods and Discussion.

Topic – light
Since You mentioned benefitial role of N 2 fixers for the whole plankton community, which is in line with Agawin et al. (2007) and other studies which You cited, it would be interesting to explicitly include a sentence about the interplay between nutrients (dominantly DIN) and light as the common controlling factors in competition between N 2 fixers and non-fixers. It is obvious that Your research went further in more details concerning the link between different types of nutrient limitation and N 2 fixation, but for me as a reader, at least a small part about the light is missing.

I suppose that since the studied area spreads within a short range of geographic latitudes, You did not consider to include the topic of light in this particular manuscript, especially because of small differences in hours of daylight over the year in such an area close to the Equator? Therefore, You focused on the topic of nutrients without mentioning light explicitly, but only indirectly via citations?

It is true that we didn't focus on the role of light in the present manuscript since it has never limited autotrophs' growth, whether or not they are diazotrophs. Intracellular quotas indeed allow us to identify at each moment the most limiting element and, for autotrophs, carbon was never limiting. In most situations, carbon intracellular quotas of autotrophs were at their maximul values, but during the bloom, since cell division accelerated, UCYN and PHYS had sometimes carbon intracellular quotas that were not maximum but phosphorous intracellular quotas were far lower.

Topic – phosphoenzymes
I suggest that You slightly improve the part about the phoshphoenyzmes that are important in usage of DOP during P-limitation (see below my suggestion). In addition, there is a fine opportunity for the inclusion of findings by Dyhrman et al. (2006) reffering to the C-P lyase enzyme in Trichodesmium, which can enable this organism to get the competitive advantage over other marine phytoplankton that do not use phosphonates that are considered to contribute even up to 25% in DOP (Dyhrman et al. 2006).

We thank E. Hrustic for this interesting suggestion to which we answer in details in the specific comment below (Page 11, Line 26).

Topic – stoichiometric ratios C:N:P for heterotrophs
I did not understand the reason for implementation of Redfield ratios for the heterotrophs since the literature supports molar C:N:P of e.g. 50:10:1. I see that later in the manuscript the rates of N and P mineralization are adjusted in the model, so You did some compensation, which seems to me rational to do. However, I think that many readers would like to see a sentence with explanation for the Redfield ratios for heterotrophs.

As suggested by recent studies, we decided to apply the same Redfield ratio , i.e. 106:16:1 to the C:N:P ratios of the upper and lower ranges of intracellular quotas for all the PFTs, including bacteria and HNF for which the C:N:P ratios of 50:10:1 were used so far. This change also brought a more consistent stoichiometry in ciliates (CIL) which were predating so far on organisms with very different stoichiometries. It is reminded however that PFT's stoichiometry is flexible (QX $\in$ [QX$^{min}$;QX$^{max}$]\$ where X $\in$ {C ; N ; P}) in the model and that the Redfield ratios are only used to link together the limits of the ranges of C, N and P intracellular quotas, (i.e. QC$^{min}$ = 106 QP$^{min}$, QC$^{max}$ = 106 QP$^{max}$,…) thereby allowing a large variety of possible C:N:P ratios in PFTs, including the 50:10:1 ratio. This has been better explained in the revised manuscript (see below).

**Technical details**
My question marks are there only to provoke Your effort to slightly improve the presentation, rather than expecting the explicit answers during the open discussion, as far as I am concerned.

Page 1:
line 8 - only differing by the presence or absence of diazotrophs
The sentence has been modified as suggested
« [..] only differing by the presence **or absence** of diazotrophs »

line 13 – which seasonal changes? It seems that Abstract would be better if you write precisely here about the main results of your research regarding those seasonal changes.
The sentence has been modified as follow :
 « [..] seasonal variations **in primary production and P availability in the upper surface waters** in simWMA [..] »

Line 22 – I feel that the logics of the sentence is inadequate. The word „although" would suggest opposing statement in the second part of the sentence, but in fact the outcome is pretty logical. The area is oligotrophic and since NH 4 and NO 3 are the main inorganic N species taken up by osmotrophs, those nutrients remain low. So, I suggest this sentence: « Since nitrate (NO 3- ) and ammonium (NH 4+ ) are the two main N sources taken up by autotrophs, their concentrations remain very low in the oligotrophic ocean being frequently growth-limiting factor in most of the open ocean.... »
We are a bit uncertain regarding the suggestion made by E. Hrustic. The word « Since » would suggest that because nitrate and ammonium are the main sources for autotrophs' N needs, their concentrations remain very low in oligotrophic ocean (meaning that this is the consumption of NO3 and NH4 that depletes N in oligotrophic areas). The reasons why oligotrophic areas have low nutrient concentrations are more because of an absence or a low external input of nutrient in the photic zone coming from the bottom of the water column by vertical mixing, by horizontal advection, atmospheric dust or river/land input.

We propose to replace « although » by « while » which could maybe enhance the comprehension of the sentence.

«  While nitrate (NO$_3^{-}$ ) and ammonium (NH$_4^{+}$ ) are the two main N sources taken up by autotrophs [..] »

Page 2:
Line 2 - Some species of prokaryotic organisms....
We thank E. Hrustic for the suggesion that we took into account in the revised manuscript
« Some **species of** prokaryotic organisms (Bacteria, Cyanobacteria, Archaea), commonly called diazotrophs or 'N2-fixers', [..] »

Line 5 – Would it be good to ammend this part with a detail about the nature of dissolved N? I mean on this „diazotrophs release dissolved inorganic and organic N...“?
We agree with E. Hrustic and propose to add the following text to the revised manuscript :
« [..] diazotrophs release **a fraction of the fixed N in the dissolved pool under the form of NH4+ and dissolved organic nitrogen (DON)** in the surface waters [..]
Lines 7-8: I suggest „since it would reduce the N limitation for the phytoplankton and thus enhance primary production in the oligotrophic regions.
Explanation: This is not "characteristic" because some oligotrophic seas are P-limited, therefore "characteristic" seems not to be associated to N exclusively. "in the oligotrophic regions" sounds a bit better at the end of the sentence. Just a suggestion.
We agree with E. Hrustic and took into account this suggestion in the revised manuscript.
« [..] it would reduce the  N limitation for the phytoplakton and thus enhance primary production **in the oligotrophic regions** »

Line 10 - bioavailability of dissolved iron (Fe) and phosphate (P).....
This modification has been made in the revised manuscript as follows :
« [..] availability of **dissolved** iron (**D**Fe) [..] »
Line 13 – It would be nice to have a reference a        t the end of this sentence since You write about classical paradigma?
We added the reference of **Zehr and Kudela (2011).**

After this sentence it would be nice to extend the Introduction by Agawin et al. (2007) and Rabouille et al. (2006), which You already cited, but an explicit note about the role of light for N 2 fixation in Your manuscript is lacking. You might briefly add here that N 2 fixation is highly dependent on the circadian clock (Rabouille et al. 2006) and that the success of non-diazotrophs and diazotrophs depend on the interplay between intensity of light and DIN concentration and the competition for those resources.

We thank E. Hrustic for this suggestion that we took into consideration. We therefore added the following sentence :
« [..] thereby calling into question the classical paradigm of the N limitation in the open ocean (Zehr et al., 2011). **Moreover, chemostat experiments have highlightened that N2 fixation activity was highly dependent on the circadian clock and that the success of non-diazotrophs and diazotrophs depend on the interplay between light intensity and DIN concentration, and the competition for those resources (Rabouillet et al,. 2006 and Agawin et al., 2007).** »

Line 22 – DIP availability if the statement is strictly for inorganic P.
Yes it is inorganic P, this has been changed

Page 3:

Line 3 – Is it ecosystem or You explored ecosystems, one with diazotrophy and one without?
We agree that ecosystemS accurates more in the context and modified the sentence as follows :
« [..] to simulate the complex ecosystem**s** observed during the OUTPACE cruise [..] ».
Line 20 – their different biogeochemical characteristics (only a lack of letter „i" in their)
thank you, this has been corrected

Page 4:
Lines 3,4 – I do not see in the manuscript where did You mention earlier anything about „ten years"?
Yes, indeed, we propose to rephrase the sentence as follow :
« Both simulations were run over ten year**, and** since a cyclic steady-state [..] »

Line 14 – „consumers" could be changed to „grazers"? However, this is less important.
We replaced consumers by grazers as follows :
« [..] three  **grazers** (zooplankton) and one decomposer [..] »

Line 25 – after „energy regulator" it seems appropriate to extend the sentence with „, being itself tightly linked to the daily light cylce (Rabouille et al. 2006). Just an example.
Yes thank you for the suggestion, this has been added to the revised manuscript as follows :
« [..] energy regulator, **being itself tightly linked to the daily light cycle (Rabouille et al. 2006)** »

Page 5:
Line 4 – Hereafter You have numerous dots after some units, these dots should be corrected.
Lines 5,6: Why did You change C:N:P from 50:10:1 for heterotrophs, which is supported by literature (Goldman and Dennett (2000), Fagerbakke et al. (1996), Chan et al. (2012), Alekseenko et al. (2014), for which You cite 50:10:1), to the Redfield ones?
As suggested by recent studies, we decided to apply the same Redfield ratio , i.e. 106:16:1 to the mean intracellular quotas of all the model PFTs, including bacteria and HNF for which the mean ratios 50:10:1 were used so far. This change also brought a more consistent stoichiometry in ciliates (CIL) which were predating so far on organisms with very different stoichiometries. . We propose to insert the following paragraph to justify this choice :

« While several studies have shown that the intracellular C:N:P ratios in heterotrophic bacteria tend to be below Redfield values as they were enriched in N and P (Bratbak, 1985, Goldman and Dennett, 2000, Vrede et al., 2002 ), more recent studies suggest that these ratios could be higher than 50:10:1 and highly variables in response to physical, chemical and physiological conditions ( Cotner et al., 2010, Martiny et al., 2013 ; Zimmerman et al. 2014 ). This led us to replace the 50:10:1 ratios used so far in the model for bacteria and HNF by the Redfield 106:16:1 ratio as for the other PFTs represented in the model. It is reminded however that the PFT's stoichiometry is flexible (QX Î $[QX^{min};QX^{max}]$\$ where X $\in$ {C ; N ; P}) in the model and that these ratios are only used to link together the limits of the ranges of C, N and P intracellular quotas, (i.e. $QC^{min} = 106\ QP^{min}$, $QC^{max} = 106\ QP^{max}$,…) thereby allowing a large variety of possible C:N:P ratios in PFTs, including the 50:10:1 ratio.»

Line 9 – Hereafter You write "phosphatase alkaline". Is it nicer to write alkaline phosphatase, or it has to be phosphatase alkaline?
We thereby changed « phosphatase alkaline » by « alkaline phosphatase » as rightly suggested.

Line 34 – According to which reference You chose the ratios 1000:100:1 ? Can You extend the sentence with a comment about grazing pressure from right to the left in these abundances ratio?
Unfortunately, we are not able to provide a specific reference regarding this ratio. However, it is

reminded that these ratios only concern the initial conditions of zooplankton abundances which have no significant influence on their dynamics thereafter simulated by the model.

Page 7:
Line 21 – "particulate carbon biomass", "particulate organic carbon" and "C biomass" are present in the manuscript. Are some of these names associated to the same variable? Can You simplify this?
« C biomass » in line 28 has been replaced by « POC »
Line 32 – Why not "over ten years" if You are able to perform model for ten years?
As the variations are the same each year, we chose to represent only 3 years to show the seasonal variations. A figure showing the 10-years of simulation has no additional interest.

Page 8:
Line 28 –maybe extending the sentence with "compared to winter mixing".
Yes thank you for this suggestion which has been taken into account.
« [..] followed by a longer stratified period from February to July, with a shallower MLD between 25 and 30 m **compared to the 70 m reached during the winter mixing.** »

Page 9:
Line 26 – exclude one of the words, "new" or "fresh" because they carry the same information.
« fresh » has been removed.

Page 10:
Line 24 – Did you provide an abbreviation explanation for PP before this point in the manuscript?
Yes in section 2.1

Page 11:
Line 7 - However, their development also requires sufficient intensity of light and other nutrients...
Thank you for this suggestion which has been took into account in the revised manuscript.
« However, their development also requires **sufficient intensity of light and** other nutrients [..] »

Line 26 –Why "or"? Is it better to write "e.g." than "or". There are other phosphoenzymes potentially used to get P from DOP, especially by Trichodesmium. I suggest "(alkaline phosphatase, nucleotidase, polyphosphatase, phosphodiesterase)".
We added those other phosphoenzymes as suggested
« [..]phosphenzymes **(alkaline phosphatase, nucleotidase, polyphosphatase, phosphodiesterase)** »
Then You could extend the discussion by "Moreover, C-P lyase found in Trichodesmium (Dyhrman et al. 2006) is another enzyme that enables this organism to use previously considered non-bioavailable fraction of DOP, i.e. phosphonates that represent circa 25% of DOP (Dyhrman et al. 2006). Taking into account this possibility, Trichodesmium thrives in the oligotrophic oceans with a great success".

We really thank E. Hrustic for his suggestion regarding the C-P lyase found in Trichodesmium by Dyhrman et al., 2006. As mentioned in their paper, phosphonates are more difficult to degrade than monophosphoesters, which commonly led to characterize phosphonates as refractory DOP. We understand that the C-P lyase found in Trichodesmium or other Cyanobacteria challenged this point of view since C-P lyase have been found to be able to hydrolyze phosphonates. However, in our model, we only consider the labile pool of DOP (i.e. monosphophoesters), and implicitly took into account the phosphoenzymes activities by decreasing the half-saturation constant for the labile DOP assimilation for all autotrophs, as mentioned in Section 4.2.2. The Dyhrman et al. (2006) results would have been of great interest if we had considered in our model a pool of semi-labile DOP but this could

be one of our future model development since we have already considered this possibility for other reasons than giving an advantage to Trichodesmium...

Page 12:
Lines 17-18 – "around the main thermocline between 100 and 500 m depth". Can You provide a figure for the vertical profiles of temperature?
The following figure represents the vertical profile of temperature during the winter mixing showing that the main thermocline is located between 100 and 700 m depth. We therefore changed the sentence as follows :
« around **the first 400 m of** the main thermocline, between 100 and 500 m depth. »

[Figure]

*Figure 1 Modeled vertical profile of Temperature during the stratified period (March 2015)*

Page 13:
Line 7 – project, Olsen et al. (2016)). Comment: add comma after "project".
Yes thank you, the coma has been added.

Page 14:
Line 6 – nutriclines depths

we replaced « depth of nutriclines » by « nutriclines depths » as suggested in the following sentence :
« [..] (ii) the higher/ lower  **nutriclines depths** characteristic of oligotrophic (WMA)/ultra-oligotrophic (WGY) states, [..] »

Lines 8-9 – Is excess DIP ideal for growth of diazotrophs if the light intensity is not suitable? Maybe "Excess DIP in N-limited surface ocean, if supported by sufficient photosynthetically active radiation, would favor the growth of diazotrophs in comparison to non-diazotrophs"?
We propose to add the following section at the end of the sentence :
« [..] intensive N2 fixation, in the WTSP region where light intensity is high enough and not limiting the diazotrophs' growth. »

Page 24:
Figure 4c – Why do You use "Trichos" if You defined the abbreviation in the Methods as TRI?
This is an error, Trichos should be TRI. This has been corrected.

References (*already cited)

*Agawin NSR, Rabouille S, Veldhuis MJW, Servatius L, Harriët M, Van Overzee MJ, Huisman J (2007) Competition and facilitation between unicellular nitrogen-fixing cyanobacteria and non-nitrogen fixing phytoplankton species. Limnology and Oceanography 52(5): 2233–2248.

Chan et al. (2012) Transcriptional changes underlying elemental stoichiometry shifts in a marine heterotrophic bacterium. Frontiers in microbiology doi: 10.3389/fmicb.2012.00159

Dyhrman TS, Chappell PD, Haley ST, Moffett JW, Orchard ED, Waterbury JB, Webb EA (2006) Phosphonate utilization by the globally important marine diazotroph Trichodesmium. Nature 439(7072): 68–71.

Fagerbakke KM, Heldal M, Norland S (1996) Content of carbon, nitrogen, oxygen, sulphur and phosphorus in native aquatic and cultured bacteria. Aquatic Microbial Ecology (10): 15–27.

Goldman JC, Dennett MR (2000) Growth of marine bacteria in batch and continuous culture under carbon and nitrogen limitation. . Limnology and Oceanography 45(4): 789-800.

*Rabouille S, Staal M, Stal LJ, Soetaert K (2006) Modeling the dynamic regulation of nitrogen fixation in the cyanobacterium Trichodesmium sp. Applied and Environmental Microbiology 72(5): 3217–3227.